# REMI: Reconstructing Episodic Memory During Internally Driven Path Planning

**Zhaoze Wang[1]**
zhaoze@seas.upenn.edu

**Genela Morris [2,3†]**
genelam@tlvmc.gov.il

**Dori Derdikman [4†]**
derdik@technion.ac.il

**Pratik Chaudhari [1†]**
pratikac@seas.upenn.edu

**Vijay Balasubramanian [5,6,7†]**
vijay@physics.upenn.edu

[1]Dept. of Electrical and Systems Eng., Univ. of Pennsylvania
[2]Tel Aviv Sourasky Medical Center
[3]Gray Faculty of Medical and Health Sciences, Tel Aviv University
[4]Rappaport Faculty of Medicine, Technion – Israel Institute of Technology
[5]Dept. of Physics, Univ. of Pennsylvania      [6]Santa Fe Institute
[7]Rudolf Peierls Centre for Theoretical Physics, University of Oxford
[†]**Equal contribution**

## Abstract

Grid cells in the medial entorhinal cortex (MEC) and place cells in the hippocampus (HC) both form spatial representations. Grid cells fire in triangular grid patterns, while place cells fire at specific locations and respond to contextual cues. How do these interacting systems support not only spatial encoding but also internally driven path planning, such as navigating to locations recalled from cues? Here, we propose a system-level theory of MEC-HC wiring that explains how grid and place cell patterns could be connected to enable cue-triggered goal retrieval, path planning, and reconstruction of sensory experience along planned routes. We suggest that place cells autoassociate sensory inputs with grid cell patterns, allowing sensory cues to trigger recall of goal-location grid patterns. We show analytically that grid-based planning permits shortcuts through unvisited locations and generalizes local transitions to long-range paths. During planning, intermediate grid states trigger place cell pattern completion, reconstructing sensory experiences along the route. Using a single-layer RNN modeling the HC-MEC loop with a planning subnetwork, we demonstrate these effects in both biologically grounded navigation simulations using RatatouGym and visually realistic navigation tasks using Habitat Sim. Codes for experiments, simulations, and vision encoder are available at [1,2,3].

## 1  Introduction

Grid and place cells in the medial entorhinal cortex (MEC) and hippocampus (HC) form complementary spatial codes for navigation [1–3]. Grid cells (GCs) in MEC fire in periodic triangular patterns as animals move through space [4–6], spanning multiple spatial scales [7–10] potentially shaped by inhibitory gradients within attractor networks [11]. Path integration theories propose that GCs track position by integrating movement [4, 12, 13], and recurrent neural networks (RNNs) trained for this task similarly develop grid-like activity [14–18]. Hippocampal place cells (HPCs) fire at specific

---

[1]Project Page: https://zhaozewang.github.io/remi

[2]RatatouGym (Simulation Suite): https://ratatougym.github.io

[3]Bottleneck MAE: https://github.com/grasp-lyrl/btnk_mae

locations [1, 2] and remap across contexts [19–23], forming sparse, orthogonal representations that minimize contextual interference [24–26]. They auto-associate sensory sequences relayed through MEC during exploration [27–30], linking related memories via continuous attractor dynamics [31–34], and emerge in RNNs that auto-encode spatial experience [21–23].

Despite extensive study of grid and place cell function [14, 16, 18, 21–23, 35, 36], it remains unclear how these representations support internally driven navigation, such as recalling goals from partial cues and planning novel routes. This ability is fundamental to flexible, goal-directed behavior, allowing animals to adapt to novel situations without extensive relearning. Evidence from rodent rest and sleep shows that hippocampal sequences replay past trajectories and generate novel ones through unvisited locations [37, 38], suggesting an underlying mechanism for offline planning.

Two complementary frameworks have been proposed to explain planning. The successor representation (SR) posits that hippocampal place cells encode expected future occupancy [39, 40], enabling flexible replanning when goals change [41, 42] and supporting generalization across related tasks through reusable predictive representations [43]. However, SR models discretize space into learned transition states, requiring experience at many locations and limiting interpolation to unvisited areas. Grid-cell models instead propose continuous, periodic codes that support vector-based navigation [44], with stable phase relationships providing context-independent spatial metrics. Yet encoding pairwise relations among locations becomes intractable as environments scale, and pure grid-based planning cannot easily capture context-specific constraints or hippocampal replay phenomena. Each framework offers a partial explanation of the problem, but how the brain integrates context-dependent memory with context-independent spatial metrics remains unresolved.

Here, we propose a unified theory of hippocampal-entorhinal circuitry that explains how internally driven navigation arises from the interaction between place and grid cells. This specific hippocampal-entorhinal wiring can enable cue-triggered goal retrieval and grid-based planning. Building on recent work showing that place cells autoassociatively encode sensory inputs [22], we propose that they also associate these inputs with GC activity. This coupling supports bidirectional pattern completion, whereby sensory cues can reactivate corresponding grid representations to guide planning, and grid states can reconstruct the expected sensory experiences along planned routes.

We verified these effects using a single RNN trained to autoencode sensory inputs, integrate velocity signals for path integration, and form autoassociative links between sensory and grid representations. Through training, the RNN developed an internal representation of the navigable space. To enable planning, we expanded the RNN's hidden layer to include an additional planner subnetwork that drives the encoder's dynamics with internally generated action sequences, enabling the system to traverse imagined paths between the current location and a recalled goal. To test whether this framework could generalize to realistic navigation, we replaced simulated sensory patterns with visual features from panoramic images in a photorealistic environment (Habitat-Sim [45–47]), encoded using a modified Masked Autoencoder (MAE) [48]. During planning, as the network traversed imagined trajectories, intermediate sensory states were decoded by the MAE into images that closely matched the expected views at corresponding locations.

Our theory predicts that (a) during planning, HC-MEC coupling will induce "look forward" sweeps in MEC spatial representation, resembling recent experimental results [49], (b) when presented with sensory cues associated with distant locations, grid cell response patterns corresponding to those locations will be reactivated, and (c) disrupting MEC-to-HC projections should impair goal-directed navigation, while disrupting HC-to-MEC feedback should reduce planning accuracy.

## 2 Method

***Conceptual Framework of the HC-MEC Loop***. We construct a conceptual framework showing how known spatial cell types could integrate into a unified system supporting both mapping and internally driven planning. Our theory builds on four established theories: **(a)** GCs arise from integrating velocity sequences for localization [4, 12–18]; **(b)** spatially modulated cells (SMCs) in MEC respond to external sensory stimuli [27] and relay sensory inputs to hippocampus; **(c)** HPCs autoassociate noisy sensory inputs from SMCs to reconstruct denoised sensory experience during recall [21, 22, 50–52]; and **(d)** while HPCs may also support planning, GCs provide a more efficient substrate for planning through their continuous, context-independent encoding [44].

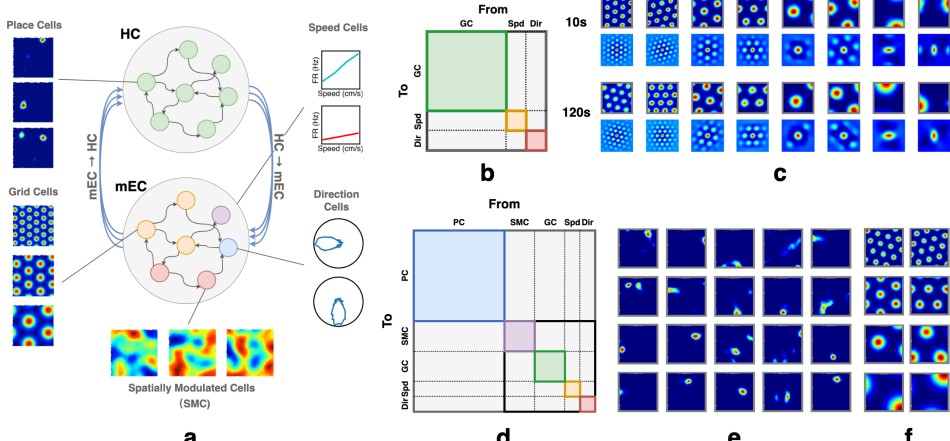

**Figure 1: (a)** RNN model of the HC-MEC loop. The top subnetwork contains HPCs, with example emergent place fields shown on the left. The bottom subnetwork includes partially supervised GCs, as well as supervised speed, head direction, and spatially modulated cells (SMCs); example grid fields shown at left. **(b)** & **(d)** Speed cells and head direction cells are denoted as *Spd* and *Dir*, respectively. Colored regions highlight within-group recurrent connectivity to indicate the partitioning of the connectivity matrix by cell groups. However, at initialization, no structural constraints are enforced. The full connectivity matrix is randomly initialized. **(b)** Illustration of the path-integration network's connectivity matrix. **(c)** The network is trained to path-integrate 5s trials and tested on 10s trials ($L = 100.50 \pm 8.49$ cm); the grid fields remain stable even in trials up to 120s ($L = 1207.89 \pm 30.98$ cm). For each subpanel (10s, 120s): top row shows firing fields; bottom row shows corresponding autocorrelograms. **(d)** Illustration of RNN connectivity matrix of full HC-MEC loop. **(e-f)** Example place fields (emergent) and grid fields in the full HC-MEC RNN model.

Since both GCs and SMCs are present in MEC and both project to hippocampus, GCs may be regarded as a special form of sensory response derived from proprioceptive signals. This anatomical arrangement suggests our core theoretical contribution, extending (c) by proposing HPCs autoassociate and pattern-complete not only SMCs but also GCs. When sensory cues are unreliable and path integration drifts over time due to noise, this association provides a natural solution for more robust localization than either system alone. This framework directly yields three key predictions: (1) partial sensory cues can reactivate the corresponding grid cell state via hippocampal autoassociation; (2) the recalled grid state can guide planning on the grid cell manifold, enabling efficient, context-independent, and generalizable path computation; (3) during planning, intermediate grid states trigger hippocampal reactivation of the corresponding sensory representations, reconstructing the expected sensory experience along the planned route.

***An RNN Model of the HC-MEC Loop***. Our theoretical framework requires an RNN that can simultaneously (1) autoencode masked and noisy sensory experiences (SMCs), (2) contain grid cells capable of path-integrating noisy movement inputs, (3) form autoassociative links between SMC and GC responses, and (4) later support planning. To achieve these, we extend RNN formulations from previous models of spatial navigation cell types [14, 16, 18, 22, 53]. Specifically, consider a standard RNN that updates its dynamics as:

$$z_{t+1} = \alpha \cdot z_t + (\mathbf{1} - \alpha) \cdot \left( \mathbf{W}^{in} u_t + \mathbf{W}^{rec} f(z_t) \right) \tag{1}$$

where $z \in \mathbb{R}^{d_z}$ is the hidden state, $u$ is the input, $\alpha$ is the forgetting rate, and $\mathbf{W}^{in}$, $\mathbf{W}^{rec}$ are the input and recurrent weight matrices. The output is given by a linear readout: $y_t = \mathbf{W}^{\text{out}} z_t \in \mathbb{R}^{d_o}$. We extend this RNN by introducing auxiliary input and output nodes, $z^I$ and $z^O$, and update as:

$$\begin{bmatrix} z_{t+1} \\ z^I_{t+1} \\ z^O_{t+1} \end{bmatrix} = \alpha \odot \begin{bmatrix} z_t \\ z^I_t \\ z^O_t \end{bmatrix} + (\mathbf{1} - \alpha) \odot \left( \begin{bmatrix} \mathbf{0} \\ u_t \\ \mathbf{0} \end{bmatrix} + \begin{bmatrix} \mathbf{W}^{rec} & \tilde{\mathbf{W}}^{in} & \mathbf{W}^{(13)} \\ \mathbf{W}^{(21)} & \mathbf{W}^{(22)} & \mathbf{W}^{(23)} \\ \tilde{\mathbf{W}}^{out} & \mathbf{W}^{(32)} & \mathbf{W}^{(33)} \end{bmatrix} f\left( \begin{bmatrix} z_t \\ z^I_t \\ z^O_t \end{bmatrix} \right) \right) \tag{2}$$

Here, $z^I$ directly integrates the input $u_t$ without a learnable projection, while $z^O$ is probed and supervised to match simulated ground-truth cell responses. This design eliminates the need for projection matrices $\mathbf{W}^{in}$ and $\mathbf{W}^{out}$. Instead, $\tilde{\mathbf{W}}^{in}$ and $\tilde{\mathbf{W}}^{out}$ act as surrogate mappings for the original projections. We set $\alpha$ as a learnable vector in $\mathbb{R}^{d_z+d_I+d_O}$ to allow different cells to have distinct forgetting rates, with $\odot$ denoting element-wise multiplication.

***Supervising Spatially Modulated Cells (SMCs)***. The SMCs are set as both input and output nodes, trained to reproduce the simulated ground truth. At each time step, SMC units receive simulated but partially occluded sensory inputs representing noisy environmental observations. After each recurrent update, the network is expected to reconstruct the corresponding noiseless sensory pattern and is penalized for deviations of the SMC subpopulation from the ground truth responses. These SMCs are assumed to primarily respond to sensory cues during physical traversal. Supervision constrains their dynamics to reflect tuning to these signals, while learned recurrent connections formed during training are intended to reflect Hebb-like updates in the brain that preserve this tuning structure.

***Supervising Grid Cells for Path-Integration***. Our theoretical model also requires grid cells that have learned to perform path integration. To achieve this, we assign two additional subpopulations within the RNN's hidden units to represent speed and allocentric head direction cells. Both subpopulations are defined as input nodes that only receive external inputs. Head direction cells are assigned preferred allocentric directions uniformly distributed over $[0, 2\pi)$ with a fixed angular tuning width, ensuring non-negative responses and a population activity that lies on a 1D ring (see Suppl. 3). We also simulate ground-truth grid cell activity maps. To train the network to perform path integration, the model is provided only with the initial grid cell responses at $t = 0$. For all subsequent timesteps, the network must infer the grid cell activity from the initial response together with the ongoing inputs from the speed and head direction cells.

This path integration component can be viewed as a modified version of the networks described in [14–16, 18]. To verify that the network indeed learned path integration, we first trained a model containing only grid, speed, and head direction cells (Figure 1b). We simulated six grid modules with spatial periods scaled by the theoretical optimal factor $\sqrt{e}$ [54]. The smallest grid spacing is set to 30 cm, defined as the distance between two firing centers of a grid cell [55]. The grid spacing to field size ratio is 3.26 [56], with firing fields modeled as Gaussian blobs with radii equal to two standard deviations. We train this model on short random trajectories (5 s) but accurately path-integrate over significantly longer trajectories (10 s, 120 s) during testing (Figure 1c).

***Emergence of Place Cell-Like Patterns During Autoassociation of SMCs and GCs***. The central idea of our HC-MEC model is that localization using sensory inputs (SMCs) or through path integration alone can each fail under noise. Associating the two representations through hippocampal place cells makes localization more robust. To model this, we introduced a subpopulation of hidden units in the RNN that are neither input nor output neurons. They do not directly receive external signals or produce outputs. These units receive input from and project to both the SMC and GC subpopulations through recurrent connections (Figure 1d). After training the full HC-MEC network, with SMC units supervised to autoencode masked sensory inputs and GC units trained to path integrate noisy movement signals, we observed the spontaneous emergence of place cell-like activity in this intermediary region. No explicit supervision or spatial constraint was imposed, and the place-like patterns emerged naturally through training, consistent with previous findings [22].

## 3 Recalling MEC Representations from Sensory Observations

We test our first prediction that associating sensory observations (SMC responses) with GC patterns enables recall of those patterns from partial sensory cues. Auto-association arises when input patterns are incomplete or degraded. To model this, we trained nine identical networks differing only in masking ratio $r_{\text{mask}}$ (0.1 to 0.9), which specifies the maximum fraction of head direction, speed, GC, and SMC inputs and initial hidden states randomly set to zero during training. Each model was trained on randomly sampled short trajectories using a fixed $r_{\text{mask}}$, with new masks generated for every trajectory and varying across time and cells. Masks were applied to both inputs and initial states of GCs, SMCs, speed cells, and head direction cells (Suppl. 5.2).

After training, we randomly selected locations in the environment and sampled the corresponding ground-truth SMC responses. Each sampled response was repeated $T$ times to form a query. During

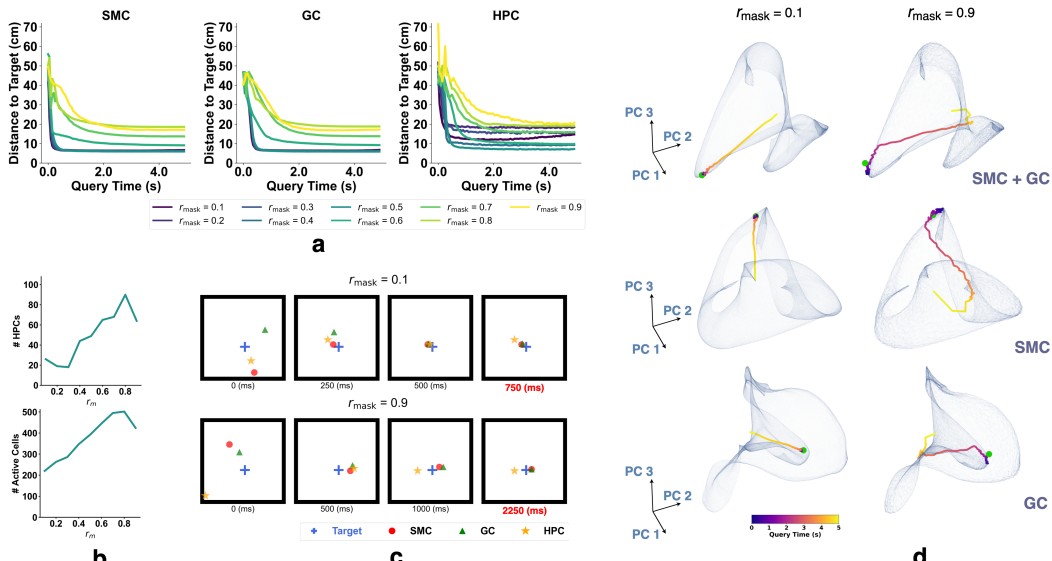

**Figure 2:** Recalling MEC representations from sensory observations with networks trained under different masking ratios $r_{\text{mask}}$. **(a–d)** Results from querying the trained network with fixed sensory input. **(a)** L2 distance between decoded and target positions using SMCs, GCs, and HPCs. **(b)** Top: number of identified HPCs vs. $r_{\text{mask}}$ (max 512). Bottom: number of active hippocampal units vs. $r_{\text{mask}}$ (max 512). **(c)** Decoded positions from SMC, GC, and HPC population responses. **(d)** Example recall trajectories for SMC, GC, and their concatenation. Semi-transparent surfaces show PCA-reduced ratemaps (extrapolated 5× for visualization) from testing. Trajectories are colored by time; green dot marks the target.

queries, the network state was initialized to zero across all hidden-layer neurons, and the query was input only to the spatial modulated cells, while responses from all cells were recorded over $T$ timesteps. At each timestep, the activity of a subpopulation of cells (e.g., SMC, GC, HPC) was decoded into a position by finding the nearest neighbor on the corresponding subpopulation ratemap aggregated during testing (Suppl.5.4). Nearest neighbor search was performed using FAISS [57, 58] (Suppl.4).

Figure 2a shows the L2 distance between decoded and ground-truth positions over time. The network was queried for 5 seconds (100 timesteps), and all models successfully recalled the goal location with high accuracy. HPCs were identified as neurons with mean firing rates above 0.01 Hz and spatial information content (SIC) above 20 (Suppl. 2). The number of HPCs increased with $r_{\text{mask}}$, and location decoding using HPCs was performed only for models with more than 10 HPCs. We observe a trade-off in which higher $r_{\text{mask}}$ leads to more HPCs and improved decoding accuracy but reduces the network's ability to recall sensory observations (Figure 2a,b).

To visualize the recall process (Figure 2c), we conducted Principal Components Analysis (PCA) on the recall trajectories. We first flattened the $L_x \times L_y \times N$ ground-truth ratemap into a $(L_x \cdot L_y) \times N$ matrix, where $L_x$ and $L_y$ are the spatial dimensions of the arena and $N$ is the number of cells in each subpopulation. PCA was then applied to reduce this matrix to $(L_x \cdot L_y) \times 3$, retaining only the first three principal components. The trajectories (colored by time), goal responses (green dot), and ratemaps collected during testing were projected into this reduced space for visualization. In Figure 2d, the recall trajectories for all subpopulations converge near the target representation, indicating successful retrieval of target SMC and GC patterns. Once near the target, the trajectories remain close and continue to circulate around it, indicating stable dynamics during the recall process.

## 4 Planning with Recalled Representations

The recall experiment shows that auto-associating spatially modulated cells (SMCs) and grid cells (GCs) through hippocampal place cells (HPCs) enables recovery of all cells' representations from

partial sensory cues. Although planning with HPCs and SMCs is feasible [40–43], their context dependence limits generalization across environments. In contrast, GC patterns encode the geometric structure of space and provide a context-independent substrate for planning. Planning on the grid manifold allows HPC auto-association to reconstruct corresponding SMC representations from intermediate GC states, removing the need to plan directly with HPCs or SMCs. We therefore propose that planning occurs on the grid cell manifold, with sensory details later recovered through hippocampus auto-associations.

## 4.1 Decoding Displacement from Grid Cells

We first revisit and reframe the formalism in [44]. Grid cells are grouped into modules based on shared spatial periods and orientations. Within each module, relative phase relationships remain stable across environments [10, 55, 59, 60]. This stability allows the population response of a grid module to be represented by a single phase variable $\phi$ [44], which is a scalar in 1D and a 2D vector in 2D environments. This variable maps the population response onto an $n$-dimensional torus [61], denoted as $\mathbb{T}^n = \mathbb{R}^n/2\pi\mathbb{Z}^n \cong [0, 2\pi)^n$, where $n \in \{1, 2\}$ is the dimension of navigable space.

Consider a 1D case with $\phi_c$ and $\phi_t$ as the phase variables of the current and target locations in some module. The phase difference is $\Delta\phi = \phi_t - \phi_c$, and since $\phi_c, \phi_t \in [0, 2\pi)$, we have $\Delta\phi \in (-2\pi, 2\pi)$. However, for vector-based navigation, we instead need $\Delta\phi^*$ such that $\phi_t = [\phi_c + \Delta\phi^*]_{2\pi}$, where $[\cdot]_{2\pi}$ is an element-wise modulo operation so that $\phi_t$ is defined on $[0, 2\pi)$. Simply using $\Delta\phi$ directly is not sufficient because multiple wrapped phase differences correspond to the same phase $\phi_t$, but different physical positions on the torus. Therefore, we restrict $\Delta\phi$ to be defined on $(-\pi, \pi)$ such that the planning mechanism always selects the shortest path on the torus that points to the target phase. The decoded displacement in physical space is then $\hat{d} \in [-\ell/2, \ell/2]$.

For 2D space, we define $\Delta\phi \in \mathbb{R}^2$ on $(-\pi, \pi)^2$ by treating the two non-collinear directions as independent 1D cases. In Figure 3a, the phase variables $\phi_c$ and $\phi_t$ correspond to two points on a 2D torus. When unwrapped into physical space, these points repeat periodically, forming an infinite lattice of candidate displacements (Figure 3b). In 2D, this yields four ($2^2$) distinct relative positions differing by integer multiples of $2\pi$ in phase space. Only the point $\phi_t^*$ lies within the principal domain $(-\pi, \pi)^2$, and the decoder selects $\Delta\phi \in (-\pi, \pi)^2$ that minimizes $\|\Delta\phi\|$, subject to $\phi_t = [\phi_c + \Delta\phi]_{2\pi}$.

## 4.2 Sequential Planning

Previous network models compute displacement vectors from GCs by directly decoding from the current $\phi_c$ and the target $\phi_t$ [44]. However, studies show that during quiescence, GCs often fire in coordinated sequences tracing out trajectories [62, 63], rather than representing single, abrupt movements toward the target. At a minimum, long-distance displacements are not directly executable and must be broken into smaller steps. What mechanism could support such sequential planning?

We first consider a simplistic planning model on a single grid module. Phase space can be discretized into $N_\phi$ bins, grouping GC responses into $N_\phi$ discrete states. Local transition rules can be learned even during random exploration, allowing the animal to encode transition probabilities between neighboring locations. These transitions can be compactly represented by a matrix $T \in \mathbb{R}^{N_\phi \times N_\phi}$, where $T_{ij}$ gives the probability of transitioning from phase $i$ to phase $j$. With this transition matrix, the animal can navigate to the target by stitching together local transitions, even without knowing long-range displacements. Specifically, suppose we construct a vector $v^{\text{plan}} \in \mathbb{R}^{N_\phi}$ with nonzero entries marking the current and target phases to represent a planning task. Multiplying $v^{\text{plan}}$ by matrix $T$ propagates the current and target phases to their neighboring phases, effectively performing a "search" over possible next steps based on known transitions.

By repeatedly applying this update, the influence of the current and target phase spreads through phase space, eventually settling on an intermediate phase that connects start and target (Figure 3c). If the animal selects the phase with the highest value after each update and renormalizes the vector, this process traces a smooth trajectory toward the target (Figure 3d). This approach can be generalized to 2D phase spaces (Figure 3e). In essence, we propose that the animal can decompose long-range planning into a sequence of local steps by encoding a transition probability over phases in a matrix. A readout mechanism can then map these phase transitions into corresponding speeds and directions and subsequently update GC activity toward the target.

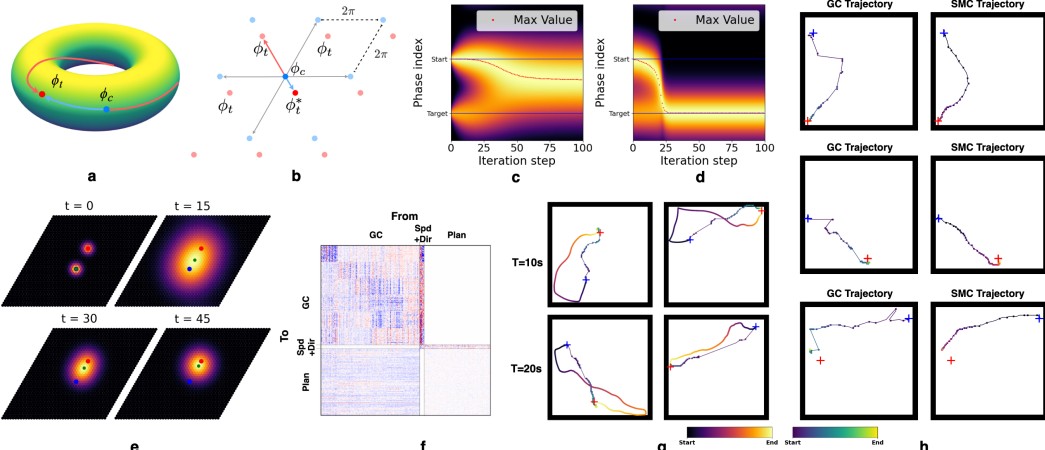

**Figure 3: (a)** The population response of a single grid module forms an $n$-dimensional torus, where multiple phase differences can connect the current and target phases. **(b)** Unwrapping phases into physical space yields $2^n$ candidate displacements; only $\phi_t^*$ lies within the principal domain $(-\pi, \pi)^n$. **(c)** A Markovian process asymptotically identifies the most likely next phase that moves closer to the target. **(d)** Renormalizing after each update produces a smooth trajectory from start to target. **(e)** Illustration of this process in 2D space. **(f)** Learned connectivity matrix of the planning RNN using only grid cells. **(g)** Planned trajectories for targets reachable within 10 and 20 seconds. Blue and red crosses mark start and target locations; the reference line shows the full trajectory for visualization. The dots represented the decoded locations. **(h)** A planning network connected to the full HC-MEC, receiving input only from GC and controlling speed and direction, drives SMC responses to update alongside GC, tracing a trajectory closely aligned with the planned GC path.

## 4.3 Combining Decoded Displacement from Multiple Scales

Our discussions of planning and decoding in sections 4.1 and 4.2 were limited to displacements within a single grid module. However, this is insufficient when $\Delta\phi$ exceeds half the module's spatial period ($\ell/2$). The authors of [44] proposed combining $\Delta\phi$ across modules before decoding displacement. We instead suggest decoding $\Delta\phi$ within each module first, then averaging the decoded displacements across modules is sufficient for planning. This procedure allows each module to update its phase using only local transition rules while still enabling the animal to plan a complete path to the target.

We start again with the 1D case. Suppose there are $m$ different scales of grid cells, with each scale $i$ having a spatial period $\ell_i$. From the smallest to the largest scale, these spatial periods are $\ell_1, \ldots, \ell_m$. The grid modules follow a fixed scaling factor $s$, which has a theoretically optimal value of $s = e$ in 1D rooms and $s = \sqrt{e}$ in 2D [54]. Thus, the spatial periods satisfy $\ell_i = \ell_0 \cdot s^i$ for $i = 1, \ldots, m$, where $\ell_0$ is a constant parameter that does not correspond to an actual grid module.

Given the ongoing debate about whether grid cells contribute to long-range planning [64, 65], we focus on mid-range planning, where distances are of the same order of magnitude as the environment size. Suppose two locations in 1D space are separated by a ground truth displacement $d \in \mathbb{R}_+$, bounded by half the largest scale ($\ell_m/2$). We can always find an index $k$ where $\ell_1/2, \ldots, \ell_k/2 \leq d < \ell_{k+1}/2 < \ldots, \ell_m/2$. Given $k$, we call scales $\ell_{k+1}, \ldots, \ell_m$ **decodable** and scales $\ell_1, \ldots, \ell_k$ **undercovered**. For undercovered scales, phase differences $\Delta\phi$ are wrapped around the torus at least one period of $(-\pi, \pi)$ and may point to the wrong direction. We thus denote the phase difference due to the undercovered scales by $Z_i$. If we predict displacement by simply averaging the decoded displacements from all grid scales, the predicted displacement

$$\hat{d} = \tfrac{\ell_0}{2\pi m} \cdot \left( \textstyle\sum_{i=1}^{k} s^i \cdot Z_i + \sum_{i=k+1}^{m} s^i \cdot \Delta\phi_i \right).$$

In 1D, the remaining distance after taking the predicted displacement is $d_{\text{next}} = d_{\text{current}} - \hat{d}$. For the predicted displacement to always move the animal closer to the target, meaning $d_{\text{next}} < d_{\text{current}}$, it

suffices that $m > k + \frac{1-s^{-k}}{s-1}$ (see Suppl. 1). This condition is trivially satisfied in 1D for $s = e$, as $\frac{1-s^{-k}}{s-1} < 1$ requiring only $m > k$. In 2D, where the optimal scaling factor $s = \sqrt{e}$, the condition tightens slightly to $m > k + 1$. Importantly, as the animal moves closer to the target, more scales become decodable, enabling increasingly accurate predictions that eventually lead to the target. In 2D, planning can be decomposed into two independent processes along non-collinear directions. Although prediction errors in 2D may lead to a suboptimal path, this deviation can be reduced by increasing the number of scales or taking smaller steps along the decoded direction, allowing the animal to gradually approach the target with higher prediction accuracy.

## 4.4 A RNN Model of Planning

***Planning Using Grid Cells Only.*** We test our ideas in an RNN framework. We first ask whether a planner subnetwork, modeled together with a GC subnetwork, can generate sequential trajectories toward the target using only grid cell representations. Accordingly, we connect a planning network to a pre-trained GC network that has already learned path integration. For each planning task, the GC subnetwork is initialized with the ground truth GC response at the start location, while the planner updates the GC state from the start to the target location's GC response by producing a sequence of feasible actions—specifically, speeds and directions. This ensures the planner generates feasible actions rather than directly imposing the target state on the GCs, while a projection from the GC region to the planning region keeps the planner informed of the current GC state. The planner additionally receives the ground truth GC response of the target location through a learnable linear projection. At each step, the planner receives $\mathbf{W}^{in}g^* + \mathbf{W}^{g\to p}g_t \in \mathbb{R}^{d_p}$, where $g^*$ and $g_t$ are the goal and current GC patterns, while $\mathbf{W}^{in}$ and $\mathbf{W}^{g\to p}$ are the input and GC-to-planner projection matrices. This combined input has the same dimension as the planner network. Conceptually, it can be interpreted as the planning vector $v^{\text{plan}}$, while the planner's recurrent matrix represents the transition matrix $T$. The resulting connectivity matrix is shown in Figure 3f.

During training, we generate random 1-second trajectories to sample pairs of current and target locations, allowing the animal to learn local transition rules. These trajectories are used only to ensure that the target location is reachable within 1 second from the current location; the trajectories themselves are not provided to the planning subnetwork. The planning subnetwork is trained to minimize the mean squared error between the current and target GC states for all timesteps.

For testing, we generate longer 10 and 20 second trajectories to define start and target locations, again without providing the full trajectories to the planner. The GC states produced during planning are decoded at each step to infer the locations the animal is virtually planned to reach. As shown in Figure 3g, the dots represent these decoded locations along the planned path, while the colored line shows the full generated trajectory for visualization and comparison. We observe that the planner generalizes from local to long-range planning and can take paths that shortcut the trajectories used to generate start and target locations. Notably, even when trained on just 128 fixed start-end pairs over 100 steps, it still successfully plans paths between locations reachable over 10 seconds.

***Training the Planning Network Together with HC-MEC Enables Reconstruction of Sensory Experiences Along Planned Paths***. We next test whether the HC-MEC loop enables the network to reconstruct sensory experiences along planned trajectories using intermediate GC states. To this end, we train the planning subnetwork together with the entire HC-MEC model that was pre-trained to encode the room during random traversal ($r_{\text{mask}} = 1.0$, see Suppl. 5.2). We fix all projections from SMCs and HPCs to the planner to zero. This ensures that the planner uses only GC representations for planning and controls HC-MEC dynamics by producing inputs to speed and direction cells. SMCs and GCs are initialized to their respective ground truth responses at the start location.

Using the same testing procedures as before, we sampled the SMC and GC responses while the planner generated paths between two locations reachable within 10 seconds. We decoded GC and SMC activity at each timestep to locations using nearest neighbor search on their respective ratemaps. We found that the decoded SMC trajectories closely followed those of the GCs, suggesting that SMC responses can be reconstructed from intermediate GC states via HPCs (see Figure 3h). Additionally, compared to the GC-only planning case, we reduced the number of GCs in the HC-MEC model to avoid an overly large network, which would make the HC-MEC training hard to converge. Although this resulted in a less smooth GC-decoded trajectory than in Figure 3g, the trajectory decoded from

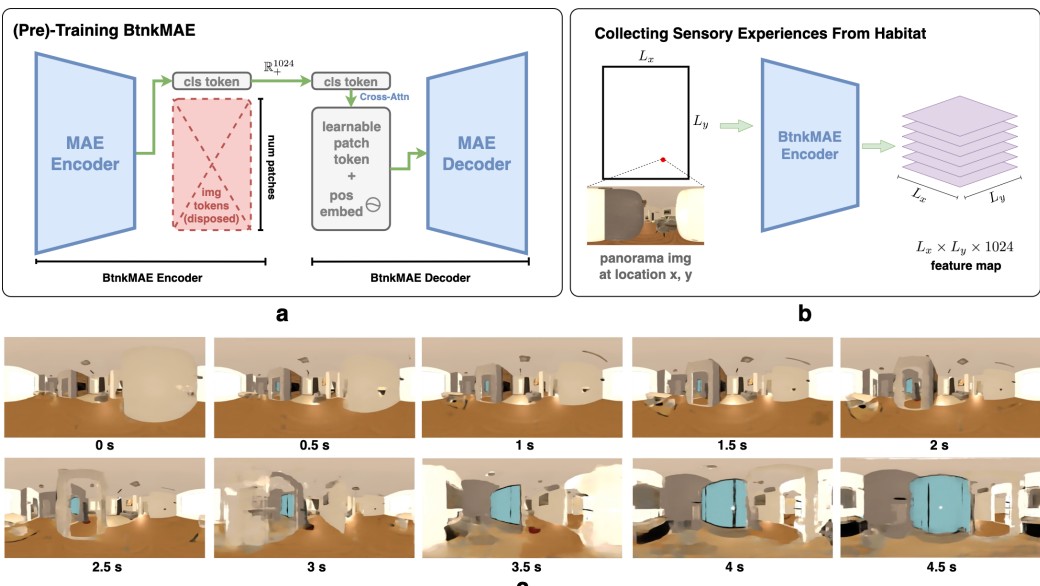

**Figure 4:** **(a)** (Pre-)training process of the BtnkMAE that uses a modification of the Masked AutoEncoder in [48]. Image features are ignored, and only the CLS token is passed to the decoder. The decoder cross-attends to the CLS token with learnable embeddings to reconstruct the image tokens such that each image can be faithfully encoded into 1024 dimensional features. See Suppl. 7. **(b)** Collection of SMC rate maps from Habitat Sim. **(c)** Decoding intermediate SMC states during path planning yields images that closely match the expected views along the planned route.

SMCs was noticeably smoother. We suggest this is due to the auto-associative role of HPCs, which use relayed GC responses to reconstruct SMCs, effectively smoothing the trajectory.

***Testing REMI for Realistic Navigation Task in Habitat Sim***. Finally, we tested whether the proposed framework generalizes to realistic navigation tasks and whether the reconstructed sensory embeddings not only correspond to the correct locations along the planned path but also retain sufficient information to recover the original sensory observations. To this end, we replaced the simulated SMC rate maps $R_{\text{smc}} \in \mathbb{R}^{L_x \times L_y \times N}$ with visual feature embeddings extracted from the Habitat Synthetic Scenes Dataset (HSSD) [66] in the photorealistic simulator Habitat-Sim [45–47]. The new SMC rate map collected in Habitat-Sim is $R_{\text{hab}} \in \mathbb{R}_+^{L_x \times L_y \times 1024}$.

To construct $R_{\text{hab}}$, we captured panoramic images $I$ at each spatial location with a fixed orientation and encoded them using a vision encoder that produced 1024-dimensional feature vectors $E(I) \in \mathbb{R}_+^{1024}$. We additionally needed a decoder $D(E(I)) \approx I$ to transform the encoded image back to the pixel space. For this purpose, we modified a pretrained Masked Autoencoder (MAE) [48] with a ViT-Large backbone, referred to as BtnkMAE. The original MAE produced image features of size $\mathbb{R}^{p \times d}$, where $p$ is the number of patches and $d$ the embedding dimension, which is unsuitable for our HC-MEC model that requires a single vector representation per image. To address this, we pretrained BtnkMAE on ImageNet-1k, retaining only a retrained CLS token after the encoder to represent each image before decoding. The decoder then employed DETR-style cross-attention [67] to reconstruct the image from this single embedding vector (Figure 4a; see also Suppl. 7 for details).

We then repeated the experiment from the previous section with an additional layer normalization to stabilize training. During planning, the HC-MEC updated the SMC region to intermediate states $z_0, \ldots, z_T$, where $z_i \in \mathbb{R}_+^{1024}$ denotes the population firing statistics of the SMC subregion. Decoding them with $D$ yielded images $I_0, \ldots, I_T$ that closely resembled the expected views at the corresponding planned locations (Figure 4c).

# 5 Discussion

Decades of theoretical and computational studies have explored how hippocampal place cells (HPCs) and grid cells (GCs) arise, addressing the question "What do they encode?" An equally important question is "Why does the brain encode?" We suggest that place and grid representations evolved to support internally driven navigation toward vital resources such as food, water, and shelter. Thus, we ask: *given their known phenomenology, how might HPCs and GCs be wired to support internally driven navigation?*

Previous studies show that HPCs are linked to memory and context but exhibit localized spatial maps that lack the relational structure required for planning, while the periodic lattice of GCs supports path planning and generalization yet exhibits weak contextual tuning that limits direct recall from sensory cues. We propose that GCs and spatially modulated cells (SMCs) form parallel localization systems encoding complementary aspects of space: GCs provide metric structure and SMCs reflect sensory observations. HPCs link these systems through auto-association to enable bidirectional pattern completion. We built a single-layer RNN containing GCs, HPCs, SMCs, and a planning subnetwork to test whether this coupling enables three key capabilities: (1) recall of GC patterns from contextual cues, (2) generalizable GC-based planning to novel routes, and (3) reconstruction of sensory experiences along planned paths, eliminating the need for direct planning on HPC or SMC manifolds.

Our model is flexible and accommodates existing theories. In REMI, HPCs must predict both GC and SMC responses at the next timestep to pattern-complete. Since HPCs are direction-agnostic, accurately reconstructing GC and SMC activity requires encoding possible state transitions. Because adjacent locations tend to have higher transition probabilities [39], our model implicitly reflects the SR framework of HPCs, in which the transition structure is embedded in the pattern-completion dynamics. This predictive nature aligns with previous theories that predictive coding in HPCs supports planning [40], but we emphasize that using GCs may be more efficient when shortcutting unvisited locations. Lastly, while we propose that HC-MEC coupling enables planning through recall and reconstruction, a parallel idea was proposed in [68], where a similar coupling was argued to support greater episodic memory capacity.

Our theory makes several testable predictions. First, sensory cues should reactivate GC activity associated with a target location, even when the animal is stationary or in a different environment. Second, during planning, HC-MEC coupling will induce "look forward" sweeps in MEC spatial representation, resembling recent experimental results [49]. Finally, if hippocampal place cells reconstruct sensory experiences during planning, disrupting MEC-to-HC projections should impair goal-directed navigation, while disrupting HC-to-MEC feedback should reduce planning accuracy by preventing animals from validating planned trajectories internally.

**Limitations:** First, existing emergence models of GCs make it difficult to precisely control GC scales and orientations [11, 17], and so to avoid the complicating analysis of the simultaneous emergence of GCs and HPCs, we supervised GC responses during training. Future work on their co-emergence could further support our proposed planning mechanism. Second, our framework does not account for boundary avoidance, which would require extending the HC-MEC model to include boundary cell representations [35, 69, 70]. Finally, our discussion of planning with GCs assumes the environment is of similar scale to the largest grid scale. One possibility is long-range planning may involve other brain regions [65], as suggested by observations that bats exhibit only local 3D grid lattices without a global structure [64]. Animals might use GCs for mid-range navigation, while global planning stitches together local displacement maps from GC activity.

## Acknowledgments and Disclosure of Funding

The study was supported by NIH CRCNS grant 1R01MH125544-01 and in part by the NSF and DoD OUSD (R&E) under Agreement PHY-2229929 (The NSF AI Institute for Artificial and Natural Intelligence). Additional support was provided by the United States–Israel Binational Science Foundation (BSF). PC was supported in part by grants from the National Science Foundation (IIS-2145164, CCF-2212519) and the Office of Naval Research (N00014-22-1-2255). VB was supported in part by the Eastman Professorship at Balliol College, Oxford.

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

# 1 Condition for Positivity of Prediction Error

Given two locations, the simplest navigation task can be framed as decoding the displacement vector between them from their corresponding grid cell representations. As suggested in Section 4.3, we propose that it is sufficient for navigation even if we first decode a displacement vector within each grid module and then combine them through simple averaging. Here, we examine what conditions are required to guarantee that the resulting averaged displacement always moves the animal closer to the target.

In the 1D case, or along one axis of a 2D case, the decoded displacement vector through simple averaging is given by:

$$\hat{d} = \frac{\ell_0}{2\pi m} \cdot \left( \sum_{i=1}^{k} s^i \cdot Z_i + \sum_{i=k+1}^{m} s^i \cdot \Delta\phi_i \right)$$

After taking this decoded displacement, the remaining distance to the target along the decoded direction is $d - \hat{d}$. To ensure the animal always moves closer to the target along this axis, it suffices to show that $d - \hat{d} < d$. Which is satisfied if $m > k + \frac{1-s^{-k}}{s-1}$.

*Proof.* Assume that all decodable scales yield the correct displacement vectors, i.e., for all $i \in \{k+1, \cdots, m\}$, we have:

$$\frac{\ell_0 \cdot s^i \cdot \Delta\phi_i}{2\pi} = d$$

Substituting into the expression for $\hat{d}$:

$$d - \hat{d} = \frac{k}{m} \cdot d - \frac{\ell_0}{m \cdot 2\pi} \cdot \left( \sum_{i=1}^{k} s^i \cdot Z_i \right)$$

And thus we require

$$d - \hat{d} < d \quad \Leftrightarrow \quad (k - m) \cdot d < \frac{\ell_0}{2\pi} \cdot \left( \sum_{i=1}^{k} s^i \cdot Z_i \right)$$

Since that $k$ is the index that delineates the decodable and undercovered scales, $(\ell_0 \cdot s^k)/2 \le d < (\ell_0 \cdot s^{k+1})/2$. The worst case occurs when $d$ takes its maximum value $(\ell_0 \cdot s^{k+1})/2$ while $Z_i$ takes its minimum value $-\pi$. Substituting these:

$$(k - m) \cdot \frac{\ell_0 \cdot s^{k+1}}{2} < \frac{\ell_0}{2\pi} \cdot \left( \sum_{i=1}^{k} s \cdot (-\pi) \right)$$

$$\Rightarrow (k - m) \cdot s^{k+1} < -\sum_{i=1}^{k} s^i = -\frac{s(s^k - 1)}{s - 1}$$

$$\Rightarrow k - m < -\frac{s(s^k - 1)}{s^k s(s - 1)} = -\frac{1 - s^{-k}}{s - 1}$$

$$\Rightarrow m > k + \frac{1 - s^{-k}}{s - 1}$$

Notice that this bound only extend the initial assumption $m > k$ by $\frac{1-s^{-k}}{s-1}$ which never exceeds 1 when $s = e$ for 1D case, and never exceeds 2 when $s = \sqrt{e}$ for 2D space. Therefore, simple averaging reliably decreases the distance to the goal if $m > k$ in 1D and $m > k + 1$ in 2D.

$\square$

## 2  Spatial Information Content

We use spatial information content (SIC) [71] to measure the extent to which a cell might be a place cell. The SIC score quantifies how much knowing the neuron's firing rate reduces uncertainty about the animal's location. The SIC is calculated as

$$I = \sum_{i}^{N} p_i \cdot \frac{r_i}{\mathbb{E}[r]} \cdot \log_2\left(\frac{r_i}{\mathbb{E}[r]}\right)$$

Where $\mathbb{E}[r]$ is the mean firing rate of the cell, $r_i$ is the firing rate at spatial bin $i$, and $p_i$ is the empirical probability of the animal being in spatial bin $i$. For all cells with a mean firing rate above 0.01Hz, we discretize their firing ratemaps into 20 pixel $\times$ 20 pixel spatial bins and compute their SIC. We define a cell to be a place cell if its SIC exceeds 20.

## 3  Simulating Spatial Navigation Cells and Random Traversal Behaviors

### 3.1  Simulating Spatial Navigation Cells

In our model, we simulated the ground truth response of MEC cell types during navigation to supervise the training. We note that firing statistics for these cells vary significantly across species, environments, and experimental setups. Moreover, many experimental studies emphasize the phenomenology of these cell types rather than their precise firing rates. Thus, we simulate each type based on its observed phenomenology and scale its firing rate using the most commonly reported values in rodents. This scaling does not affect network robustness or alter the conclusions presented in the main paper. However, the relative magnitudes of different cell types can influence training dynamics. To mitigate this, we additionally employ a unitless loss function that ensures all supervised and partially supervised units are equally emphasized in the loss (see Suppl. 5.3.3).

**Spatial Modulated Cells**: We generate spatially modulated cells following the method in [22]. Notably, the simulated SMCs resemble the cue cells described in [27]. To construct them, we first generate Gaussian white noise across all locations in the environment. A 2D Gaussian filter is then applied to produce spatially smoothed firing rate maps.

Formally, let $\mathcal{A} \subset \mathbb{R}^2$ denote the spatial environment, discretized into a grid of size $W \times H$. For each neuron $i$ and location $\mathbf{x} \in \mathcal{A}$, the initial response is sampled as i.i.d. Gaussian white noise: $\epsilon_i(\mathbf{x}) \sim \mathcal{N}(0, 1)$. Each noise map $\epsilon_i$ is then smoothed via 2D convolution with an isotropic Gaussian kernel $G_{\sigma_i}$, where $\sigma_i$ represents the spatial tuning width of cell $i$. The raw cell response is then given by:

$$R_i^{\text{raw}} = \epsilon_i * G_{\sigma_i}$$

where $*$ denotes the 2D convolution. The spatial width $\sigma_i$ is sampled independently for each cell using $\mathcal{N}(12\text{cm}, 3\text{cm})$. Finally, the response of each cell is normalized using min-max normalization:

$$R_i = \frac{R_i^{\text{raw}} - \min(R_i^{\text{raw}})}{\max(R_i^{\text{raw}}) - \min(R_i^{\text{raw}})}$$

The SMCs used in our experiments model the sensory-related responses and non-grid cells in the MEC. Cue cells reported in [27] typically exhibit maximum firing rates ranging from 0–20Hz, but show lower firing rates at most locations distant from the cue. Non-grid cells reported in [72] generally have peak firing rates between 0–15Hz. To align with these experimental observations, we scale all simulated SMCs to have a maximum firing rate of 15Hz.

**Grid Cells**: To simulate grid cells, we generate each module independently. For a module with a given spatial scale $\ell$, we define two non-collinear basis vectors

$$\mathbf{b}_1 = \begin{bmatrix} \ell \\ 0 \end{bmatrix} \quad \text{and} \quad \mathbf{b}_2 = \begin{bmatrix} \ell/2 \\ \ell\sqrt{3}/2 \end{bmatrix}$$

These vectors generate a regular triangular lattice:

$$\mathcal{C} = \{n\mathbf{b}_1 + m\mathbf{b}_2 \mid n, m \in \mathbb{Z}\}$$

For each module, we randomly pick its relative orientation with respect to the spatial environment by selecting a random $\theta \in [0, \pi/3)$ which is used to rotate the lattice:

$$\mathbf{R}^\theta = \begin{bmatrix} \cos\theta & -\sin\theta \\ \sin\theta & \cos\theta \end{bmatrix}, \quad \mathcal{C}_\theta = \{\mathbf{R}^\theta \mathbf{c} \mid \mathbf{c} \in \mathcal{C}\}$$

Within each module, individual cells are assigned unique spatial phase offsets. These offsets $\psi_i$ are sampled from a equilateral triangle with vertices

$$V_1 = \mathbf{R}^\theta \begin{bmatrix} 0 \\ 0 \end{bmatrix} \quad V_2 = \mathbf{R}^\theta \begin{bmatrix} 0 \\ -\ell/2 \end{bmatrix} \quad V_3 = \mathbf{R}^\theta \begin{bmatrix} \ell\sqrt{3}/2 \\ -\ell/2 \end{bmatrix}.$$

We sample phase offsets for grid cells within each module by drawing vectors uniformly from the triangular region using the triangle reflection method. Since the resulting grid patterns are wrapped around the lattice, this is functionally equivalent to sampling uniformly from the full parallelogram. The firing centers for cell $i$ in a given module are then given by:

$$\mathcal{C}_i = \{\mathbf{c}_i^* + \psi_i \mid \mathbf{c}_i^* \in \mathcal{C}_\theta\}$$

Finally, the raw firing rate map for cell $i$ is generated by placing isotropic Gaussian bumps centered at each location in $\mathcal{C}_i$:

$$R_i^{\text{raw}} = \sum_{\mathbf{c}_i \in \mathcal{C}_i} \exp\left(-\frac{\|\mathbf{x} - \mathbf{c}_i\|^2}{2\sigma_{\text{grid}}^2}\right), \quad \mathbf{x} \in \mathcal{A}$$

where $\sigma_{\text{grid}}$ is the spatial tuning width of each bump and is computed as: $2\sigma_{\text{grid}} = \ell/r$ with $r = 3.26$ following the grid spacing to field size ratio reported in [56]. Experimental studies have reported that rodent grid cells typically exhibit maximum firing rates in the range of 0–15Hz [7, 55], though most observed values are below 10Hz. Accordingly, we scale the generated grid cells to have a maximum firing rate of 10Hz.

**Speed Cells**: Many cells in the MEC respond to speed, including grid cells, head direction cells, conjunctive cells, and uncategorized cells [73, 74]. These cells may show saturating, non-monotonic, or even negatively correlated responses to movement speed. To maintain simplicity in our model, we represent speed cells as units that respond exclusively and linearly to the animal's movement speed. Specifically, we simulate these cells with a linear firing rate tuning based on the norm of the displacement vector $\vec{d}$ at each time step, which reflects the animal's instantaneous speed.

Additionally, many reported speed cells exhibit high baseline firing rates [73, 74]. To avoid introducing an additional parameter, we set all speed cells' tuning to start from zero—i.e., their firing rate is 0Hz when the animal is stationary. To introduce variability across cells, each speed cell is assigned a scaling factor $s_i$ sampled from $\mathcal{N}(0.2, 0.05)$ Hz/(cm/s), allowing cells to fire at different rates for the same speed input. Given that the simulated agent has a mean speed of 10cm/s (Suppl. 3.2), the average firing rate of speed cells is approximately 2Hz. We chose a lower mean firing rate than observed in rodents, as we did not include a base firing rate for the speed cells. However, this scaling allows cells with the strongest speed tuning to reach firing rates up to 80Hz, matching the peak rates reported in [74] when the agent moves at its fastest speed.

All cells follow the same linear speed tuning function:

$$R_i(\vec{d}) = \frac{s_i \cdot \|\vec{d}\|}{dt}$$

where $dt$ is the simulation time resolution (in seconds), $\|\vec{d}\|$ is the displacement magnitude over that interval, and $s_i$ modulates the response sensitivity of each cell.

**Direction Cells**: We define direction cells based on the movement direction of the animal, rather than its body or head orientation. During initialization, each cell is assigned a preferred allocentric direction drawn uniformly at random from the interval $[0, 2\pi)$. At any time during the animal's movement, we extract its absolute displacement vector $\vec{d}$. From this vector, we compute the angular difference $\Delta\theta$ between the allocentric movement direction and the $i$-th cell's preferred direction. The movement direction can be derived from the displacement vector $\vec{d}$. Each neuron responds according

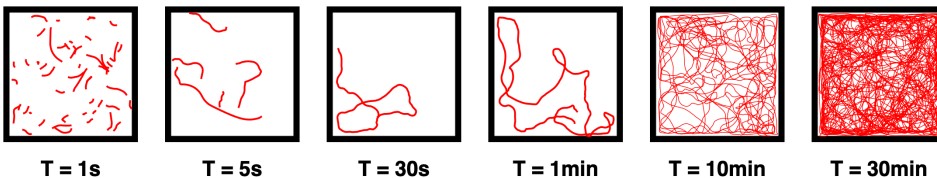

**Figure S1:** Example generated trajectories of varying lengths. From left to right: 50 trajectories of 1s each, 5 trajectories of 5s, followed by single trajectories of 30s, 1min, 10min, and 30min.

to a wrapped Gaussian tuning curve:

$$R_i(\theta) = \exp\left(-\frac{[\Delta\theta_i]_{2\pi}^2}{2\sigma_{\text{dir}}^2}\right)$$

where $[\Delta\theta_i]_{2\pi}$ denotes the angular difference wrapped into the interval $[0, 2\pi)$. $\sigma_{\text{dir}}$ denotes the tuning width (standard deviation) of the angular response curve. We set $\sigma_{\text{dir}} = 1$ rad to reduce the number of direction cells needed to span the full angular space $[0, 2\pi)$, thereby decreasing the size of the RNN to improve training efficiency. Given that many head direction cells in the MEC are conjunctive with grid and speed cells [75], we set the mean firing rate of direction cells to 2Hz to match the typical firing rates of speed cells.

We acknowledge that this simulation of the direction cell may only serve as a simplified model. However, in our model, direction cells serve only to provide input to grid cells for path integration, and we have verified that the precise tuning width and magnitude do not affect the planning performance or change the conclusion of the main text. Our findings could be further validated by future studies employing more biologically realistic simulation methods.

## 3.2   Simulating Random Traversal Behaviors

We train our HC-MEC model using simulated trajectories of random traversal, during which we sample masked observations of displacement and sensory input. The network is trained to reconstruct the unmasked values of speed, direction, SMC, and grid cell responses at all timepoints. Trajectories are simulated within a $100 \times 100\text{cm}^2$ arena, discretized at 1 cm resolution per spatial bin. The agent's movement is designed to approximate realistic rodent behavior by incorporating random traversal with momentum.

At each timestep, the simulated agent's displacement vector $\vec{d}$ is determined by its current velocity magnitude, movement direction, and a stochastic drift component. The base velocity $v$ is sampled from a log-normal distribution with mean $\mu_{\text{spd}} = 10$ and standard deviation $\sigma_{\text{spd}} = 10$ (in cm/s), such that the agent spends most of its time moving below 10 cm/s but can reach up to 50 cm/s. The velocity is converted to displacement by dividing by the simulation time resolution $dt$, and re-sampled at each timestep with a small probability $p_{\text{spd}}$ to introduce variability.

To simulate movement with momentum, we add a drift component that perturbs the agent's displacement. The drift vector is computed by sampling a random direction, scaling it by the current velocity and a drift coefficient $c_{\text{drift}}$ that determines the drifting speed. Drift direction is resampled at each step with a small probability $p_{\text{dir}}$ to simulate the animal switch their traversal direction. The drift is added to the direction-based displacement, allowing the agent to move in directions slightly offset from its previous heading. This results in smooth trajectories that preserve recent movement while enabling gradual turns.

To prevent frequent collisions with the environment boundary, a soft boundary-avoidance mechanism is applied. When the agent is within $d_{\text{avoid}}$ pixels of a wall and its perpendicular distance to the wall is decreasing, an angular adjustment is applied to its direction. This correction is proportional to proximity and only engages when the agent is actively moving toward the wall. We set $dt = 0.01$ sec/timestep, $p_{\text{spd}} = 0.02$, $c_{\text{drift}} = 0.05$, $p_{\text{dir}} = 0.15$, and $d_{\text{avoid}} = 10$ pixels. These values were chosen to produce trajectories that qualitatively match rodent traversal (see Fig S1).

# 4 Decoding Location from Population Activity

In the main text, we decode the population vector to locations in both the recall task (Section 3) and the planning task (Section 4.4). Here we present the method we used for such decoding. Given a population vector $\mathbf{r} \in \mathbb{R}^N$ at a given timestep, we decode it into a location estimate by performing nearest neighbor search over a set of rate maps corresponding to the subpopulation of cells corresponding to $\mathbf{r}$. These rate maps may come from our simulated ground-truth responses or be aggregated from the network's activity during testing (see Suppl. 5.4).

Formally, let $\mathbf{r} = [r_1, r_2, \cdots, r_N]$ be the population response of a subpopulation of $N$ cells at a given timestep, and let $M \in \mathbb{R}^{P \times N}$ be the flattened rate maps, where each row $m_p$ corresponds to the population response at the $p$-th spatial bin. Here, $P = W \times H$ is the total number of discretized spatial bins in the environment $\mathcal{A}$. The decoding process is to find the index $p^*$ that minimizes the Euclidean distance between $\mathbf{r}$ and $m_p$:

$$p^* = \arg\min_p \|\mathbf{r} - m_p\|_2$$

To efficiently implement this decoding, we use the FAISS library [57, 58]. Specifically, we employ the `IndexIVFFlat` structure, which first clusters the rows $\{m_p\}_{p=1}^P$ into $k$ clusters using $k$-means. Each vector $m_p$ is then assigned to its nearest centroid, creating an inverted index that maps each cluster to the set of vectors it contains.

At query time, the input vector $\mathbf{r}$ is first compared to all centroids to find the `n_probe` closest clusters. The search is then restricted to the vectors assigned to these clusters. Finally, the nearest neighbor among them is returned, and its index $p^*$ is mapped back to the corresponding spatial coordinate $\mathbf{x}^*$. For all experiments, we set the `n_clusters` for the k-means to 100 and `n_probe` to 10.

# 5 Training and Testing of the HC-MEC Model

As described in Section 2.1, our HC-MEC model is a single-layer RNN composed of multiple sub-networks. We train two versions of this model: (1) **GC-only** variant, which includes only grid cells and along with speed and direction cells; and (2) the full **HC-MEC** loop model, which includes both MEC and hippocampal (HC) subpopulations.

In both cases, we simulate ground-truth responses for supervised and partially supervised cells using the method described in Suppl. 3. The number of simulated cells matches exactly the number of corresponding units in the RNN hidden layer. That is, if we simulate $N_g$ grid cells, we assign precisely $N_g$ hidden units in the RNN to represent them, and (partially)-supervise these units with the corresponding ground-truth activity. Additional details on this partial supervision are provided in Suppl. 5.1.

For both models, we use six scales of grid cells, with the initial spatial scale set to 30cm. Subsequent scales follow the theoretically optimal scaling factor $s = \sqrt{e}$ [54]. The grid spacing to field size ratio is fixed at 3.26 [56]. Each scale comprises 48 grid cells, and the spatial phase offsets for each cell within a module are independently sampled from the corresponding equilateral triangle (Suppl. 3) with side length equals to the spatial scale of the module. In addition, both the **GC-only** and **HC-MEC** models include 32 speed cells and 32 direction cells.

The **HC-MEC** model additionally includes spatially modulated cells (SMCs) in the MEC subpopulation and hippocampal place cells (HPCs) in the HC subpopulation. We include 256 SMCs to approximately match the number of grid cells. These SMCs are designed to reflect responses to sensory experience and are trained with supervision as described in Suppl. 3. We also include 512 HPCs, matching approximately the total number of grid cells and SMCs (N = 256 + 288). These cells only receive input from and project to the MEC subpopulation through recurrent connections, and thus do not receive any external input. We note that, differing from [22], we did not apply a firing rate constraint on the HPCs, but still observed the emergence of place cell-like responses.

In total, unless otherwise specified, our **GC-only** model comprises 352 hidden units ($48 \times 6$ grid cells + 32 speed cells + 32 direction cells), while the **HC-MEC** model comprises 1120 hidden units (288 grid cells + 32 speed cells + 32 direction cells + 256 SMCs + 512 HPCs).

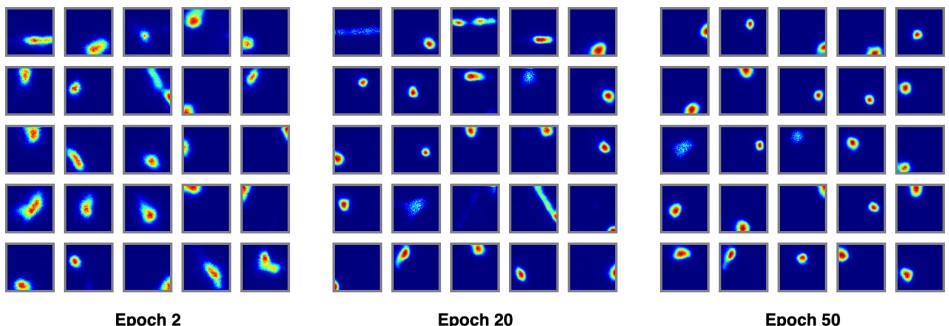

**Figure S2:** Example place fields emerging over training. Shown are place fields from epoch 2, 20, and 50. As training progresses, the fields become increasingly specific and spatially refined.

## 5.1 Supervising Grid Cells

As described in Section 2.1, we partially supervise the grid cell subpopulation of the RNN using simulated ground-truth responses. Let $\mathbf{z}_t \in \mathbb{R}^{N_g}$ denote the hidden state of the RNN units modeling grid cells at time $t$. During training, the HC-MEC model is trained on short random traversal trajectories. Along each trajectory, we sample the ground-truth grid cell responses from the simulated ratemaps at the corresponding location and denote these as $\{\mathbf{r}_t^g\}_{t=0}^T$. At the start of each trajectory, we initialize the grid cell units with the ground-truth response at the starting location, i.e., $\mathbf{z}_0 = \mathbf{r}_0^g$.

From time step $t = 1$ to $T$, the grid cell hidden states $\mathbf{z}_t$ are updated solely through recurrent projections between the grid cell subpopulation and the speed and direction cells. They do not receive any external inputs. After the RNN processes the speed and direction inputs over the entire trajectory, we collect the hidden states of the grid cell subpopulation $\{\mathbf{z}_t\}_{t=1}^T$ and minimize their deviation from the corresponding ground-truth responses $\{\mathbf{r}_t^g\}_{t=1}^T$. The training loss function is described in Suppl. 5.3.3.

We choose to partially supervise the grid cells due to their critical role in the subsequent training of the planning network. This supervision allows the model to learn stable grid cell patterns, reducing the risk of instability propagating into later stages of training. While this partial supervision does not reflect fully unsupervised emergence, it can still be interpreted as a biologically plausible scenario in which grid cells and place cells iteratively refine each other's representations. Experimentally, place cell firing fields appear before grid cells but become more spatially specific as grid cell patterns stabilize, potentially due to feedforward projections from grid cells to place cells [76, 77].

We observe a similar phenomenon during training. Even with partial supervision, grid cells tend to emerge after place cells. This may be because auto-associating spatially modulated sensory representations is easier than learning the more structured path-integration task hypothesized to be performed by grid cells. After the grid cells' pattern stabilizes, we also observe that place cell firing fields become more refined (Fig. S2), consistent with experimental findings [3, 77–79].

## 5.2 HC-MEC Training Task

During navigation, animals may use two different types of sensory information to help localize themselves: (1) sensory input from the environment to directly observe their location; and (2) displacement information from the previous location to infer the current location and potentially reconstruct the expected sensory observation. We posit that the first type of information is reflected by the weakly spatially modulated cells (SMCs) in the MEC, while the second type is reflected by grid cells and emerges through path integration.

However, as we previously argued in Section 2.1, both types of information are subject to failure during navigation. Decades of research have revealed the strong pattern completion capabilities of hippocampal place cells. We thus hypothesize that hippocampal place cells may help reconstruct one type of representation from the other through auto-associative mechanisms.

To test this hypothesis, we simulate random traversal trajectories and train our **HC-MEC** model with masked sensory inputs. The network is tasked with reconstructing the ground-truth responses of all MEC subpopulations from simulation, given only the masked sensory input along the trajectory. Specifically, the supervised units—SMCs, speed cells, and direction cells—receive masked inputs to simulate noisy sensory perception. We additionally mask the ground-truth responses used to initialize both supervised and partially supervised units at $t = 0$, so that the network dynamics also begin from imperfect internal states.

To simulate partial or noisy observations, we apply masking on a per-trajectory and per-cell basis. For a given trajectory, we sample the ground-truth responses along the path to form a matrix $\mathbf{R} = [\mathbf{r}_0 \cdots \mathbf{r}_T]^\top \in \mathbb{R}^{T \times N}$, where $N$ is the number of cells in the sampled subpopulation and $\mathbf{r}_t$ is the ground-truth population response at time $t$. The masking ratio $r_{\text{mask}}$ defines the maximum fraction of sensory and movement-related inputs, as well as initial hidden states, that are randomly zeroed during training to simulate partial or degraded observations. We generate a binary mask $\mathbf{M} \in \{0, 1\}^{T \times N}$ by thresholding a matrix of random values drawn uniformly from $[0, 1]$, such that approximately $100 \times r_{\text{mask}}$ percent of the entries are set to zero. The final masked response is then obtained by elementwise multiplication $\tilde{\mathbf{R}} = \mathbf{R} \odot \mathbf{M}$.

During training, we sample multiple trajectories to form a batch, with each trajectory potentially having a different masking ratio and masked positions. For both the HC-MEC model used in the recall task (Section 2) and the one pre-trained for the planning task (Section 4.4), masking ratios for SMCs and other inputs are sampled independently from the interval $[0, r_{\text{mask}}]$. Specifically, for each trial, we sample:

$$m_{\text{SMC}}, m_{\text{other}} \sim \mathcal{U}(0, r_{\text{mask}})$$

We use $m_{\text{SMC}}$ to generate the mask for SMC cells, and $m_{\text{other}}$ to independently generate masks for grid cells, speed cells, and direction cells. This allows the model to encounter a wide range of noise conditions during training—for example, scenarios where sensory inputs are unreliable but displacement-related cues are available, and vice versa.

## 5.3 RNN Implementation

### 5.3.1 Initialization

Our HC-MEC model includes multiple sub-regions. However, we aim to model these sub-regions without imposing explicit assumptions about their connectivity, as the precise connectivity—particularly the functional connectivity between the hippocampus (HC) and medial entorhinal cortex (MEC)—remains unknown. Modeling these sub-regions with multiple hidden layers would implicitly enforce a unidirectional flow of information: the second layer would receive input from the first but would not project back. Specifically, a multi-layer RNN with two recurrent layers can be represented by a block-structured recurrent weight matrix:

$$\mathbf{W} = \begin{bmatrix} \mathbf{W}^{11} & \mathbf{W}^{12} \\ \mathbf{W}^{21} & \mathbf{W}^{22} \end{bmatrix}$$

Where $\mathbf{W}^{11}$ and $\mathbf{W}^{22}$ are the recurrent weight matrices of the first and second recurrent layers, respectively, and $\mathbf{W}^{12}$ is the projection weights from the first to the second layer. In typical multi-layer RNN setups, $\mathbf{W}^{12} = \mathbf{0}$, meaning that the second sub-region does not send information back to the first. This structure generalizes to deeper RNNs, where only the diagonal blocks $\mathbf{W}^{ii}$ and the blocks directly below the diagonal $\mathbf{W}^{(i+1)i}$ are non-zero.

Therefore, we model the HC-MEC system as a large single-layer RNN, such that all sub-blocks are initialized as non-zero, and their precise connectivity is learned during training and entirely defined by the task. To initialize this block-structured weight matrix, we first initialize each subregion independently as a single-layer RNN with a defined weight matrix but no active dynamics. Suppose we are modeling $N_r$ subregions, and each subregion $i$ contains $d_i$ hidden units. We initialize the recurrent weights within each subregion using a uniform distribution:

$$\mathbf{W}^{ii} \sim \mathcal{U}\left(-1/\sqrt{d_i}, 1/\sqrt{d_i}\right)$$

For each off-diagonal block $\mathbf{W}^{ij}$, corresponding to projections from subregion $j$ to subregion $i$, we similarly initialize:

$$\mathbf{W}^{ij} \sim \mathcal{U}\left(-1/\sqrt{d_j}, 1/\sqrt{d_j}\right)$$

Note that the initialization bound is determined by the size of the source subregion $j$, consistent with standard practices for stabilizing the variance of the incoming signals. Once initialized, all sub-blocks are copied into their respective locations within a full recurrent weight matrix:

$$\mathbf{W}_{\text{HC-MEC}} = \begin{bmatrix} \mathbf{W}^{11} & \cdots & \mathbf{W}^{1N} \\ \vdots & \ddots & \vdots \\ \mathbf{W}^{N_r 1} & \cdots & \mathbf{W}^{N_r N_r} \end{bmatrix}$$

with total size $\sum_{i=1}^{N_r} d_i \times \sum_{i=1}^{N_r} d_i$.

Additionally, as described in main text, both input and output neurons are already modeled within the HC-MEC model. As a result, no additional input or output projections are required. Input neurons in this assembled RNN directly integrate external signals from the simulation, while the states of output neurons are directly probed out during training.

### 5.3.2 Parameters

For models used in the recall and planning tasks, we use the following parameters:

**Table 1:** Shared parameters for **GC-only** and **HC-MEC** models

| Parameter | Value | Description |
|---|---|---|
| $N_{\text{speed}}$ | 32 | Number of speed cells |
| $N_{\text{direction}}$ | 32 | Number of direction cells |
| $N_{\text{grid}}$ (per module) | 48 | Number of grid cells per module |
| n_modules | 6 | Number of spatial scales (modules) |
| initial_scale | 30 cm | Spatial period of the smallest module |
| spatial_scale_factor | $\sqrt{e}$ | Scale ratio between adjacent modules |
| $r_{\text{grid/field}}$ | 3.26 | Grid spacing to field size ratio |
| activation | ReLU | Activation function of the hidden units |
| $dt$ | 0.05 s | Time res for both cell simulation and RNN, 1s = 20 bins |
| $\alpha_{\text{init}}$ | 0.2 | Initial forgetting rate of the cells |
| learn_alpha | True | Whether the forgetting rate is learned during training |
| optimizer | AdamW | Optimizer |
| learning_rate | 0.001 | Learning rate |
| batch_size | 128 | Batch size, each batch corresponds to a single trajectory |
| n_epochs | 50 | Number of training epochs |
| n_steps | 1000 | Number of steps per trajectory |
| $T_{\text{trajectory}}$ | 2 s | Duration of each training trajectory |

In Section 4.4 of the main text, we noted that the **GC-Only** model used for planning uses $N_{\text{grid}} = 128$, i.e., each module comprised 128 grid cells. This larger population improves planning accuracy, likely due to denser coverage of the space. However, for the full **HC-MEC** model used in planning, we reverted to $N_{\text{grid}} = 48$, consistent with our default configuration. As discussed in the main text, the auto-associative dynamics from place cells help smooth the trajectory, even when the decoded trajectory from grid cells is imperfect.

### 5.3.3 Loss Function

We use a unitless MSE loss for all supervised and partially supervised units such that all supervised cell types are equally emphasized. For each trajectory, we generate ground-truth responses by sampling the corresponding region's simulated ratemaps at the trajectory's locations, resulting in $\{\mathbf{r}_t\}_{t=0}^{T}$. After the RNN processes the full trajectory, we extract the hidden states of the relevant units to obtain $\{\mathbf{z}_t\}_{t=0}^{T}$. To ensure that all loss terms are optimized equally and are not influenced by the scale or variability of individual cells, we perform per-cell normalization of the responses. For each region

**Table 2:** Additional parameters for **HC-MEC** model

| Parameter | Value | Description |
|---|---|---|
| $N_{\text{SMC}}$ | 256 | Number of spatially modulated cells |
| $\sigma_{\text{SMC}}$ | $\mathcal{N}(12\text{cm}, 3\text{cm})$ | Width of Gaussian smoothing to generate SMCs, which controls the spatial sensitivity of SMCs (see Suppl. 3) |
| $N_{\text{HPC}}$ | 512 | Number of hippocampal place cells |

$i$, we compute the mean $\boldsymbol{\mu}^i \in \mathbb{R}^{d_i}$ and standard deviation $\boldsymbol{\sigma}^i \in \mathbb{R}^{d_i}$ of the ground-truth responses across the time and batch dimensions, independently for each cell. The responses are then normalized elementwise:

$$\hat{\mathbf{r}}_t^i = \frac{\mathbf{r}_t^i - \boldsymbol{\mu}^i}{\boldsymbol{\sigma}^i}, \quad \hat{\mathbf{z}}_t^i = \frac{\mathbf{z}_t^i - \boldsymbol{\mu}^i}{\boldsymbol{\sigma}^i}$$

The total loss across all regions is computed as the mean squared error between the normalized responses:

$$\mathcal{L} = \frac{1}{T} \sum_{i=1}^{N_r} \sum_{t=1}^{T} \lambda_i \left\| \hat{\mathbf{z}}_t^i - \hat{\mathbf{r}}_t^i \right\|_2^2$$

where $\lambda$ controls the relative weights of each cell types. We set the $\lambda = 10$ for both grid cells and SMCs, while $\lambda = 1$ for velocity and direction cells. These relative weights are set as animals may emphasize their reconstruction of the sensory experience and relative location, rather than their precise speed and direction during their spatial traversal.

### 5.4 Constructing Ratemaps During Testing

After training, we test the agent using a procedure similar to training to estimate the firing statistics of hidden units at different spatial locations and construct their ratemaps. Specifically, we pause weight updates and generate random traversal trajectories. The supervised and partially supervised units are initialized with masked ground-truth responses, and the supervised units continue to receive masked ground-truth inputs at each timestep. We record the hidden unit activity of the RNN at every timestep and aggregate their average activity at each spatial location.

Let $\mathbf{z}_t \in \mathbb{R}^d$ be the hidden state or a subpopulation of hidden states of the RNN at time $t$, where $d$ is the number of hidden units. For each unit $i$, the ratemap value at location $\mathbf{x}$ is computed as:

$$R_i(\mathbf{x}) = \begin{cases} \frac{1}{N(\mathbf{x})} \sum_{t:\mathbf{x}_t=\mathbf{x}} \mathbf{z}_t^i & \text{if } N(\mathbf{x}) > 0 \\ \texttt{NaN} & \text{otherwise} \end{cases}$$

where $N(\mathbf{x})$ is the number of times location $\mathbf{x}$ was visited during testing. We set the trajectory length to $T = 5\text{s}$ (250 time steps), $\texttt{batch\_size} = 512$, and $\texttt{n\_batches} = 200$. We perform this extensive testing to ensure that the firing statistics of all units are well estimated. Each spatial location is visited on average $1008.13 \pm 268.24$ times (mean ± standard deviation).

## 6 Recall Task

In this paper, we have posited that the auto-association of place cells may trigger the reconstruction of grid cell representations given sensory observations. To test whether such reconstruction is possible, we trained nine different models, each with a fixed masking ratio $r_{\text{mask}}$ ($m_r$ in the main text) ranging from 0.1 to 0.9. Following the procedure described in Suppl. 5.2, each model was trained with the same masking ratio applied to all subregions and across all trials, but the masking positions were generated independently for each trial. That is, for each trial, $100 \times r_{\text{mask}}$ of the entries were occluded. The mask was applied both to the ground-truth responses used to initialize the network and to the subsequent inputs.

After training, we tested recall by randomly selecting a position in the arena and sampling the ground-truth response of the SMC subpopulation to represent the sensory cues. This sampled response was then repeated over $T = 10\text{s}$ (200 time steps) to form a constant input query. Unlike

training, the initial state of the network was set to zero across all units, such that the network dynamics evolved solely based on the queried sensory input.

# 7 Testing on Realistic Navigation

## 7.1 Integrating REMI into Visually Realistic Navigation Task

Having demonstrated the feasibility of our REMI framework in a simulated navigation environment where trajectories generated synthetic cell responses (Gaussian random fields as SMCs), we next tested its generalization to realistic visual navigation. To this end, we let the agent explore a visually realistic environment while converting visual observations into simulated cell responses. Importantly, since our framework predicts that intermediate GC activity during planning can drive HPCs to reconstruct intermediate sensory experiences, we further tested whether these reconstructed sensory states could be decoded back into images that match the expected views along the planned path.

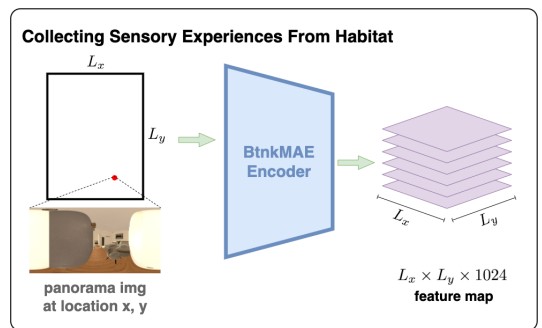

**Figure S3:** Converting $L_x \times L_y \times 512 \times 1024$ panorama image tensor into $R_{\text{hab}} \in \mathbb{R}^{L_x \times L_y \times d}$.

We used the Habitat Synthetic Scene Dataset (HSSD) [66] within the Habitat-Sim simulator [45–47]. To convert visually realistic scenes into cell responses, we first captured panoramic images at fixed orientations across all spatial locations, forming a tensor of shape $L_x \times L_y \times 512 \times 1024$, where $512 \times 1024$ is the image resolution and $L_x \times L_y$ defines the environment's spatial grid. A vision encoder $E(\cdot)$ then maps each panoramic image $I$ to a low-dimensional feature vector $E(I) \in \mathbb{R}^d$, where $d$ represents the number of sensory cells (SMCs) responding to visual signals. During planning, a decoder $D(\cdot)$ reconstructs images from SMC states such that $D(E(I)) \approx I$. This vision encoder will thus convert the $L_x \times L_y \times 512 \times 1024$ panorama image tensor into $R_{\text{hab}} \in \mathbb{R}^{L_x \times L_y \times d}$, which can then be used to replace the original $R_{\text{smc}}$. This encoding–decoding pair enables (1) compact representation of visual observations during navigation and (2) visualization of planned trajectories by decoding intermediate sensory states back into images.

## 7.2 Pre-Training Bottleneck Masked AutoEncoder

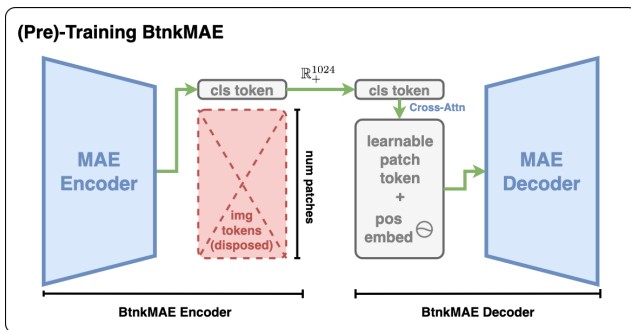

**Figure S4: Illustration of our BtnkMAE.** The original image patch tokens $\mathbf{x}_{1:p}$ are discarded after passing through the BtnkMAE encoder, which retains only the CLS token $\mathbf{x}_{\text{cls}}$ as a compact visual representation. For the REMI experiment, a ReLU activation was added after the encoder to enforce non-negative firing rates consistent with biological realism. The decoder then employs DETR-style cross-attention between a set of learnable query embeddings $\mathbf{q}_{1:p}$ and $\mathbf{x}_{\text{cls}}$ to reconstruct the discarded patch tokens $\hat{\mathbf{x}}_{1:p}$.

To construct a generalized vision encoder consisting of both an encoder $E(\cdot)$ and a decoder $D(\cdot)$, we used the Masked Autoencoder (MAE) framework [48]. The standard MAE with a Vision Transformer (ViT) backbone encodes each image into a set of patchwise features of shape $p \times d$, where $p$ is the number of image patches and $d$ is the embedding dimension. In our HC–MEC model, however, each location-specific panoramic image must be represented by a single $d$-dimensional vector, requiring compression from $p \times d$ to $d$. Simply pooling patch features would discard spatial structure, whereas retaining all patch tokens would produce inputs too high-dimensional for our recurrent architecture.

**Bottleneck MAE (BtnkMAE)**. To address this, we developed BtnkMAE, a modified MAE with a ViT backbone that compresses visual information into a single learned representation. In the original MAE, the encoder outputs features $[\mathbf{x}_{\text{cls}}, \mathbf{x}_{1:p}] \in \mathbb{R}^{(p+1) \times d}$, where $\mathbf{x}_{\text{cls}}$ is a null classification token and $\mathbf{x}_{1:p}$ are patch embeddings. In BtnkMAE, we retrain the encoder such that it discards $\mathbf{x}_{1:p}$ and passes only $\mathbf{x}_{\text{cls}}$ to the decoder. To reconstruct the discarded patch tokens, we employ DETR-style cross-attention [67], where learned query embeddings

$$\hat{\mathbf{x}}_{1:p} = f_{\text{cross\_attn}}(q_{1:p}, \mathbf{x}_{\text{cls}}) \approx \mathbf{x}_{1:p}.$$

Finally, $\hat{\mathbf{x}}_{1:p}$ are fed into the MAE decoder to reconstruct the full image.

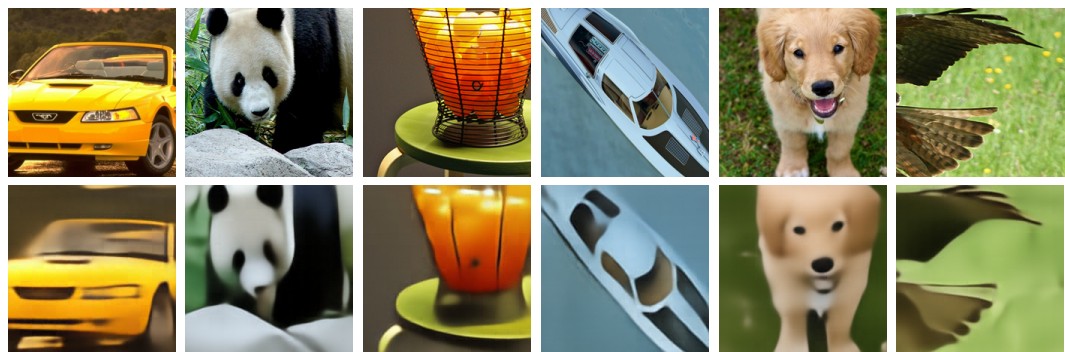

**Figure S5: First row**: original ImageNet 1k images. **Second row**: reconstructed ImageNet 1k images.

**Pretraining BtnkMAE on ImageNet 1k**. We pretrained BtnkMAE on ImageNet 1k for 100 epochs to learn a general visual compression rule, namely to distill image information into a single CLS token while maintaining decodability. We initialized BtnkMAE with pretrained weights from the original MAE, loading only parameters from the shared architecture. The CLS token weights were discarded and reinitialized, and each learnable patch embedding was augmented with an additional positional embedding to encode spatial relationships among patches. The encoder learning rate was set to 0.1 times that of the decoder to better preserve its pretrained structure. Unlike the original MAE, which is trained with random masking, BtnkMAE was trained on full images to learn compressed representations rather than reconstructing masked inputs. During training, we computed two losses, one identical to the original MAE objective that bypassed the bottleneck and another that passed through the bottleneck, allowing the model to learn compression while retaining general visual feature representations. Example views of original and reconstructed images are shown in Figure S5.

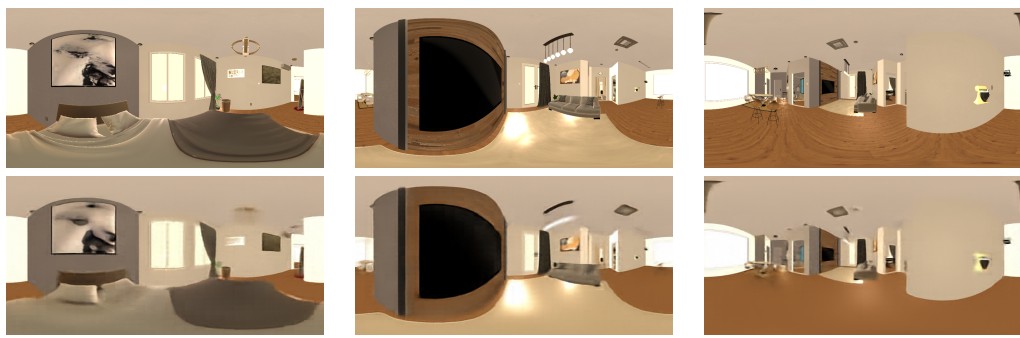

**Figure S6: First row**: original panoramic images. **Second row**: reconstructed panoramic images.

**Fine-Tuning BtnkMAE in Habitat Sim Scene**. We then fine-tuned the model on panoramic images from Habitat Sim environment. These panoramic images, originally with a resolution of $512 \times 1024$, were resized to $512 \times 512$ using bicubic interpolation to match standard ViT input dimensions. During pretraining on ImageNet-1k, the model was trained on images of size $224 \times 224$. To accommodate

the higher resolution, we recomputed fixed sinusoidal cosine positional embeddings corresponding to the $512 \times 512$ input size and fine-tuned the model specifically on panoramic views from the target navigation scenes. The original and reconstructed image pairs are shown in Figure S6.

### 7.3 Testing REMI on Visually Realistic Navigation Task

Finally, after fine-tuning BtnkMAE on the Habitat Sim scenes, we froze the encoder to convert each location-specific panoramic image into a 1024-dimensional feature vector. For an environment of shape $L_x \times L_y$, this produced a ratemap tensor $R_{\text{hab}} \in \mathbb{R}^{L_x \times L_y \times 1024}$, which replaced the SMC ratemap tensor $R_{\text{smc}}$. We then repeated the HC–MEC training described in Section 2 and the planning experiments in Section 4.4. During planning, the HC–MEC network updated the SMC region to intermediate states $z_0, \ldots, z_T$. We collected these states, each representing $z_i \in \mathbb{R}_+^{1024}$, corresponding to the agent's internal planning or virtual navigation through the environment. These states were passed to the trained and frozen BtnkMAE decoder, which successfully reconstructed images resembling the expected visual views along the planned paths.

## 8 Validation of Robustness

While our experiments rely on supervised grid cell that learns path-integration, we sought to confirm that our framework does not critically depend on ideal grid cells with idealized hexagonal lattices or noiseless inputs. To test this, we conducted additional experiments that systematically distorted the simulated ground truth grid cell representations. We trained 18 HC-MEC and corresponding planner models, 10 to examine the impact of noise and 8 to assess geometric distortions. Each model was evaluated on 4,096 planning trials. A trial was considered successful if the agent's final position was within 10 cm of the goal, and the success rate was computed across all trials. On average, the goal was located $57.54 \pm 25.53$ cm from the starting position.

For the noise experiments, we added post-activation Gaussian noise (standard deviation 0.1 to 1.0) to all hidden units of the HC-MEC at each time step during training. No noise was added at test time (planning phase).

Here, we report results based on decoding spatial locations from SMC states using their ground truth ratemaps, though results are consistent when decoding from GCs. In all cases, the mean decoded distance to the goal remained below 10 cm in a $100\,\text{cm} \times 100\,\text{cm}$ environment, and success rates remained high across noise levels. These findings indicate that the planner performs robustly even with noisy or distorted grid cell representations.

**Table 3:** Performance of the HC-MEC planner under varying noise levels.

| Noise (std) | Distance to Goal (cm) | Success Rate | Detour Ratio |
|:-----------:|:---------------------:|:------------:|:------------:|
| 0.1 | $2.20 \pm 5.89$ | 0.99 | 1.66 |
| 0.2 | $2.35 \pm 5.68$ | 0.99 | 2.00 |
| 0.3 | $1.82 \pm 5.09$ | 0.99 | 1.88 |
| 0.4 | $3.75 \pm 8.84$ | 0.98 | 2.10 |
| 0.5 | $2.62 \pm 3.43$ | 0.98 | 2.36 |
| 0.6 | $3.12 \pm 3.72$ | 0.97 | 2.33 |
| 0.7 | $3.31 \pm 5.66$ | 0.99 | 2.55 |
| 0.8 | $2.98 \pm 2.78$ | 0.99 | 2.60 |
| 0.9 | $3.00 \pm 3.99$ | 0.98 | 2.31 |
| 1.0 | $5.30 \pm 6.98$ | 0.91 | 2.70 |

Next, we added noise to the ground-truth GC signal that is used at the first time step during planning (the initial GC state).

**Table 4:** Planning performance of REMI under varying noise levels.

| Noise (std) | Distance to Goal (cm) | Success Rate | Detour Ratio |
|---|---|---|---|
| 0.1 | $2.45 \pm 2.49$ | 0.99 | 2.16 |
| 0.2 | $2.40 \pm 4.37$ | 0.99 | 1.91 |
| 0.3 | $1.89 \pm 5.41$ | 0.99 | 1.98 |
| 0.4 | $4.10 \pm 7.66$ | 0.96 | 2.54 |
| 0.5 | $4.36 \pm 5.01$ | 0.95 | 3.28 |
| 0.6 | $4.05 \pm 5.48$ | 0.96 | 2.44 |
| 0.7 | $4.01 \pm 6.70$ | 0.97 | 2.53 |
| 0.8 | $4.79 \pm 6.00$ | 0.91 | 2.63 |
| 0.9 | $5.54 \pm 8.43$ | 0.91 | 2.67 |
| 1.0 | $7.72 \pm 13.24$ | 0.85 | 2.49 |

Additionally, we trained another 8 models without noise but with sheared grid fields. We incrementally adjust the angle between the two co-linear axes of the generated grid cells used for supervision, and repeated the test above on the 8 trained models.

**Table 5:** Planning performance of REMI under geometric distortion of grid lattices.

| Shearing Angle | Residual to Goal (cm) | Success Rate | Detour Ratio |
|---|---|---|---|
| 40 | $9.53 \pm 16.39$ | 0.86 | 2.45 |
| 45 | $7.88 \pm 17.66$ | 0.91 | 2.10 |
| 50 | $7.40 \pm 18.18$ | 0.91 | 2.12 |
| 55 | $5.48 \pm 11.38$ | 0.92 | 2.38 |
| 60 | $7.66 \pm 16.29$ | 0.87 | 2.26 |
| 65 | $2.47 \pm 7.13$ | 0.98 | 2.04 |
| 70 | $5.13 \pm 10.22$ | 0.91 | 2.44 |
| 75 | $6.38 \pm 14.26$ | 0.91 | 2.12 |
| 80 | $4.46 \pm 11.31$ | 0.94 | 1.91 |

