# OpenReview forum: "REMI: Reconstructing Episodic Memory During Internally Driven Path Planning"
_NeurIPS.cc/2025/Conference — NeurIPS 2025 poster_

### Official Review · Reviewer_KTLJ · 2025-06-30

**Clarity:** 3
**Significance:** 2
**Originality:** 2
**Rating:** 5
**Confidence:** 4

**Summary:**

This paper presents a computational model, REMI, that proposes a novel, system-level functional architecture for the hippocampus-medial entorhinal cortex (HC-MEC) circuit in the service of goal-directed navigation. The central hypothesis is that hippocampal place cells (HPCs) act as an auto-associative memory system. This system is posited to link two distinct but complementary streams of information processed in the MEC: context-rich, sensory-driven representations from spatially modulated cells (SMCs) and context-invariant, metric spatial representations from grid cells (GCs). This framework seeks to explain the functional utility of maintaining these two seemingly parallel spatial coding schemes.

**Questions:**

The authors must perform extensive robustness analyses. They should systematically add noise, geometric distortions (e.g., shear, compression), and phase shifts to the "ground truth" GC signals during both training and testing. The key question is whether the planning mechanism degrades gracefully or fails catastrophically. This would, at a minimum, characterize the fragility of their approach.

The introduction and discussion sections must be expanded to provide a detailed, nuanced, and fair comparison between the REMI and SR frameworks. This discussion should clearly articulate the architectural and functional differences and, most importantly, highlight specific experimental phenomena or predictions where the two models diverge.

All experiments must be repeated multiple times (e.g., 5-10 runs) using different random seeds for all stochastic elements of the model and data generation. All figures presenting quantitative results (e.g., Figure 2a, L2 distance over time) must include error bars (e.g., shaded regions representing standard deviation or standard error of the mean) to visualize the variance across runs.

The conceptual leap from the discrete-state Markovian planning model (visualized in Fig. 3c-e) to the continuous-state RNN implementation needs to be better articulated. The paper claims the RNN's recurrent matrix represents the transition matrix T, but this is an analogy. A more detailed explanation of how the continuous dynamics of the planner RNN approximate the iterative, search-like process would significantly improve the clarity of Section 4.

**Ethical Concerns:**

["NO or VERY MINOR ethics concerns only"]

**Final Justification:**

The paper has been substantially improved and all my concerns were addressed. I have therefore increased my score to accept.

**Limitations:**

If the authors address the questions above, then yes.

**Quality:**

3

**Strengths And Weaknesses:**

The foremost strength of this paper is its ambition and success in proposing a unified, system-level theory that integrates disparate cognitive functions—sensory perception, spatial representation, memory, and planning—into a single, coherent computational framework.

Perhaps the most novel and impactful contribution is the concept of "reconstructing experience" during planning. This provides a concrete, mechanistic model for complex cognitive phenomena like hippocampal preplay and mental simulation. The model suggests that planning is not merely an abstract calculation of a trajectory in a state space, but an embodied, imaginative process where the agent can internally "visualize" the sensory consequences of a potential path.

The technical implementation of the REMI model is well-conceived and appropriate for the questions being investigated. The authors develop a modified single-layer RNN that integrates input and output nodes directly into the recurrent dynamics (Equation 2). This architectural choice elegantly eliminates the need for separate input and output projection matrices, simplifying the model while enabling the joint simulation of the HC-MEC loop and the planner within a single, unified structure.

Despite its conceptual strengths, the paper is built on a methodological flaw: the decision to use partial supervision to train the GC subnetwork. The authors justify this choice by stating their primary focus is on planning rather than emergence. While this is a pragmatic decision, it fundamentally undermines the strength, generality, and biological plausibility of their conclusions. The success of the entire planning mechanism may be critically, and perhaps entirely, contingent on having access to this idealized spatial representation. It is highly uncertain whether the proposed planning algorithm would function effectively, or at all, with the noisy, distorted, and imperfect grid representations that would likely arise from a more realistic unsupervised learning process or those observed in biological recordings. This makes the model's impressive performance potentially brittle and, worse, an artifact of the unrealistic assistance provided by the supervision.

The authors' treatment of SR is cursory and dismissive. They claim that SR-based models require visiting all locations in the environment and therefore cannot explain the ability to take shortcuts through unvisited areas. This is an oversimplification. The flexibility of SR depends heavily on the nature of the state representation and the policy generalization used; some formulations of SR can indeed support novel trajectories, e.g. see https://arxiv.org/abs/2107.08426, https://papers.nips.cc/paper_files/paper/2017/hash/350db081a661525235354dd3e19b8c05-Abstract.html. The paper makes no attempt to formally or empirically compare REMI and SR. For instance, could an SR model, where states are defined by SMC patterns, also learn to reconstruct intermediate states during planning? What are the unique, falsifiable predictions of REMI that would allow it to be experimentally distinguished from SR?

Another weakness that compromises the paper is the lack of statistical validation for the experimental results. The main quantitative and qualitative results, presented in Figures 2 and 3, are shown without any error bars, confidence intervals, or indication that the simulations were run multiple times with different random initializations.

---

> ### Author Rebuttal · Authors · 2025-07-31
>
> Thank you for the insightful comments. We are glad you see this as an ambitious and unified theory for sensory perception, spatial representation, memory, and planning. We agree that a key novelty of this work is that planning involves explicit reconstruction of experiences and an embodied, imaginative process where the agent visualizes the sensory consequences of a potential path.
>
> We appreciate your comments. Our responses are below.
>
> > **On partial supervision of grid cells**
>
> Our core contributions do not depend on co-emergence of grid cells or idealized grid responses.
>
> First, we are showing that if place cells (PCs) auto-associate grid cells and spatially modulated cells (SMCs), the agent can use sensory cues to recall GC patterns. Both recall and reconstruction of sensory experiences during planning work, as long as GC and SMC patterns are spatially stable.
>
> We will revise Section 4 to include a brief mathematical argument showing that planning with grid cells does not require idealized responses. Within a single grid scale, the displacement vector can be decoded even if the field is sheared or compressed (in 1D, this changes scale $\ell$; in 2D, it reduces to two non-colinear 1D components). Larger displacements across multiple grid-scales in distorted grid-fields can be pieced together by the decoding process. Decoding is also robust with sub-optimal grid-scales. The equation on Line 276 can be generalized as $$ \hat{d} = \frac{1}{2\pi m} \cdot \left( \sum_{i=1}^{k} \ell_{i} \cdot Z_i + \sum_{j=k+1}^{m} \ell_{j} \cdot \Delta \phi_i \right) $$ So long as there are sufficiently decodable scales, the predicted displacement reliably moves the agent toward the target.
>
> Unlike place cells, _**grid cell patterns might not be learned but are believed to be self-organized by a pre-wired network**_. For instance, deleting an NMDA receptor subunit gene disrupts grid patterns without affecting other spatial cells (Gil et al., Nat Neurosci., 2018), suggesting a genetic basis for grid-like activity. Theoretical works also suggest that grid patterns emerge from dynamical attractor network (Fuhs & Touretzky, J. Neurosci., 2006; Burak & Fiete, PLoS Comput Biol, 2009; Gardner et al., Nature, 2022). Our supervised GC module could be aslo replaced by existing models (Cueva & Wei, ICLR, 2018; Kang & Balasubramanian, eLife, 2019; Sorscher et al., NeurIPS, 2019) without affecting our conclusions. We use grid cells simply as spatial pattern generator and ask: if such patterns exist, could the animal use them for recall and planning?
>
> > **.... to the "ground truth" GC signals during both training and testing**
>
> To clarify, at training time, the GC sub-network receives the ground-truth GC signal as input at the first time-step, path-integrates the hidden state forward, and receives a loss that measures the deviation from the ground-truth GC signal at each time-step. At test time, it receives the ground-truth only at the first step. We discuss new experiments based on your suggestion in the next section.
>
> > **Statistical validation for the experimental results**
> > **perform extensive robustness analyses by systematically adding noise, geometric distortions (e.g., shear, compression) ...**
>
> We conducted additional experiments to systematically distort the simulated ground truth grid cells. We trained 18 HCMEC and corresponding planner models (10 to evaluate the impact of noise and 8 to assess geometric distortions). Each model was evaluated on 4,096 trials. A trial is successful if the final position is within 10 cm of the goal. On average, the goal is 57.54 ± 25.53 cm away from the start.
>
> For the noise experiments, we added post-activation Gaussian noise (standard deviation 0.1 to 1.0) to all hidden units of the HCMEC at each time step during training. No noise was added at test time (planning phase).
>
> Due to space constraints, we report results based on decoding from SMCs, though we verified that results are consistent when decoding from GCs. In all cases, the mean distance to the goal remains below 10 cm (in a 100 cm × 100 cm environment), and success rates remain high across noise levels. This suggests that the planner can effectively operate even with noisy GC representations.
>
> | Noise (std) | Distance to Goal (cm) | Success Rate | Detour Ratio |
> | - | - | - | - |
> | 0.1 | 2.20 ± 5.89 | 0.99 | 1.66 |
> | 0.2 | 2.35 ± 5.68 | 0.99 | 2.00 |
> | 0.3 | 1.82 ± 5.09 | 0.99 | 1.88 |
> | 0.4 | 3.75 ± 8.84 | 0.98 | 2.10 |
> | 0.5 | 2.62 ± 3.43 | 0.98 | 2.36 |
> | 0.6 | 3.12 ± 3.72 | 0.97 | 2.33 |
> | 0.7 | 3.31 ± 5.66 | 0.99 | 2.55 |
> | 0.8 | 2.98 ± 2.78 | 0.99 | 2.60 |
> | 0.9 | 3.00 ± 3.99 | 0.98 | 2.31 |
> | 1.0 | 5.30 ± 6.98 | 0.91 | 2.70 |
>
> Next, we added noise to the ground-truth GC signal that is used at the first time step during planning.
>
> | Noise (std) | Distance to Goal (cm) | Success Rate | Detour Ratio |
> | - | - | - | - |
> | 0.1 | 2.45 ± 2.49 | 0.99 | 2.16 |
> | 0.2 | 2.40 ± 4.37 | 0.99 | 1.91 |
> | 0.3 | 1.89 ± 5.41 | 0.99 | 1.98 |
> | 0.4 | 4.10 ± 7.66 | 0.96 | 2.54 |
> | 0.5 | 4.36 ± 5.01 | 0.95 | 3.28 |
> | 0.6 | 4.05 ± 5.48 | 0.96 | 2.44 |
> | 0.7 | 4.01 ± 6.70 | 0.97 | 2.53 |
> | 0.8 | 4.79 ± 6.00 | 0.91 | 2.63 |
> | 0.9 | 5.54 ± 8.43 | 0.91 | 2.67 |
> | 1.0 | 7.72 ± 13.24 | 0.85 | 2.49 |
>
> Additionally, we trained another 8 models without noise but with sheared grid fields.
>
> | Shearing Angle | Residual to Goal (cm) | Success Rate | Detour Ratio |
> | - | - | - | - |
> | 40 | 9.53 ± 16.39 | 0.86 | 2.45 |
> | 45 | 7.88 ± 17.66 | 0.91 | 2.10 |
> | 50 | 7.40 ± 18.18 | 0.91 | 2.12 |
> | 55 | 5.48 ± 11.38 | 0.92 | 2.38 |
> | 60 | 7.66 ± 16.29 | 0.87 | 2.26 |
> | 65 | 2.47 ± 7.13 | 0.98 | 2.04 |
> | 70 | 5.13 ± 10.22 | 0.91 | 2.44 |
> | 75 | 6.38 ± 14.26 | 0.91 | 2.12 |
> | 80 | 4.46 ± 11.31 | 0.94 | 1.91 |
>
> We also trained 10 models with 60-degree grid cell lattices, without noise, using different random seeds. The mean distance to the goal was 3.14 ± 2.30 cm.
>
> > **Experiments on real datasets**: We highlight an additional contribution: as suggested by reviewer b2n6, we repeated our experiments in a visually realistic environment (Habitat-Sim).
>
> In our model, SMCs represent sensory observations. While the main text uses simulated sensory signals, here we replace them with features derived from visual observations in the Habitat Synthetic Scenes Dataset (HSSD). The agent captures images during exploration, which are processed by a vision encoder to generate SMC signals. We follow the same training procedure as in Section 4.4 and find the model successfully reconstructs visual features along the path.
>
> The vision encoder produces features that can be decoded back into images, allowing visualization of scenes along the planned path. Intermediate SMC states generated during planning decode into images that closely match expected views at their corresponding planned locations. Key implementation details are as follows:
>
> - We discretize the environment into an $X_w \times X_h$ grid and capture $W \times H$ allocentric panorama images at each location.
>
> - We fine-tuned a visual encoder (a modified Masked AutoEncoder; He et al., 2021) on images from this simulation. It encodes images into compact features that can be decoded back: $D(E(I)) \approx I$, where $E: \mathbb{R}^{W \times H} \to \mathbb{R}^{512}$ and $D: \mathbb{R}^{512} \to \mathbb{R}^{W \times H}$.
>
> - Each image is encoded into a 512-dim vector used as the SMC response, enabling training of HCMEC and the planner as before.
>
> - During planning, the HCMEC updates the SMC region to intermediate states ${z_0, \dots, z_T}$, where $z_i \in \mathbb{R}^{512}$. Decoding these yields a sequence of images that closely resemble the expected views at planned locations.
>
> > **Successor Representation**
>
> We will remove the sentence “SR-based models require visiting all locations…” on Line 57.
>
> Introduction
> “Successor representation (SR) frameworks (e.g., Stachenfeld et al., Nature Neurosci., 2017; Levenstein et al., bioRxiv, 2024) propose that hippocampal place cells (HPCs) encode expected future state occupancy under a given policy, enabling flexible behavior across related tasks. These models provide compelling accounts of generalization in structured environments and task transfer (Barreto et al., NeurIPS, 2017; Abdolsheh et al., ICML, 2021). However, most SR models often emphasize discrete state spaces represented by HPCs, which exhibit strong contextual tuning. In contrast, grid cells in the medial entorhinal cortex (MEC) display continuous and periodic spatial structure that supports vector-based navigation. It is less context-specific and potentially allows greater generalization across environments. If grid cells encode spatial transition likelihoods, as suggested by the stability of their phase relationships, they may therefore offer a complementary substrate for planning.”
>
> Discussion
> “Our framework provides a complementary perspective to existing SR models. In REMI, HPCs must predict both GC and SMC responses at the next timestep in order to pattern-complete. Since HPCs are direction-agnostic, accurately reconstructing GC and SMC activity requires encoding possible state transitions. Because adjacent locations tend to have higher transition probability (Stachenfeld et al., 2017), our model implicitly reflects the SR framework of HPCs, where transition structure is embedded in the pattern-completion dynamics. This may be validated experimentally: selectively disrupting HPC-to-MEC projections may degrade planning under REMI, but not SR frameworks.”
>
> >**Conceptual leap from the discrete-state Markovian planning model**
>
> Our discussion of the Markov chain serves primarily as a conceptual framework to explain how the RNN might generalize from short to long trajectories, and why such generalization depends on the periodic grid structure. It is challenging to directly verify whether the RNN instantiates the specific Markov chain we describe. We will revise the paragraph on Line 238 to clarify this intent.

---

> > ### Comment · Area_Chair_gtyS · 2025-08-05
> >
> > Dear Reviewer,
> >
> > The authors have already responded to your initial questions. As the deadline for the reviewer-author interaction session is approaching on August 6th, please begin addressing any further concerns or questions you may have. If you have no additional queries, kindly update your rating and submit your final decision.
> >
> > Thank you for your valuable contributions to NeurIPS.
> >
> > Best, AC

---

> > ### Comment · Reviewer_KTLJ · 2025-08-07
> >
> > I wish to thank the authors for addressing all my concerns and adding experiments and validations. I will increase my score to accept.

---

> > > ### Author Response · Authors · 2025-08-07
> > >
> > > Thank you very much for your thoughtful and constructive suggestions. We are glad to hear that we’ve addressed your concerns.
> > >
> > > Best,
> > > The Authors

---

### Official Review · Reviewer_b4bC · 2025-07-01

**Clarity:** 2
**Significance:** 3
**Originality:** 2
**Rating:** 3
**Confidence:** 3

**Summary:**

The authors present a single-layer RNN model of the MEC-HPC circuit, incorporating speed cells, direction cells, spatially modulated cells , and grid cells, with place cells emerging during training. The model learns path integration and can retrieve spatial locations from masked sensory cues. Additionally, the authors introduce a planning submodule—implemented both as a theoretically motivated algorithm and a trained RNN—which generates navigation trajectories between current and goal locations.

**Questions:**

- Could the authors clarify the use of the term “partially supervised” about grid cells? Based on the description, it appears that grid cell activity is fully supervised against ground truth responses.
- Could the authors elaborate on the path integration mechanism? Specifically, how do speed and direction cells project to grid cells? Do any neurons with conjunctive tuning to location, speed, and direction emerge during training?
- Why is the SMC trained separately on masked sensory inputs? My understanding is that auto-association should be learned during memory formation (i.e., path integration), and masking should be applied only during testing to evaluate recall.
- Has the planner RNN been analyzed or dissected to confirm that it implements the proposed planning mechanism described in Sections 4.1–4.3?

**Ethical Concerns:**

["NO or VERY MINOR ethics concerns only"]

**Final Justification:**

The topic addressed in this work is interesting, and the authors have demonstrated the model’s capability for goal-directed path planning as well as the important role of GCs. However, I still have two major concerns:
As noted in the rebuttal, the reconstruction of sensory experience from GC patterns is described as “a core contribution we aim to emphasize in this paper.” To my understanding, similar questions have been addressed in prior work, for example, Chandra, S. et al., Nature, 2025.
In their further reply, the authors acknowledged that their work extends Chandra, S. et al., “emphasizing how the same underlying wiring logic might also support complex goal-directed planning.” However, in the subsequent planning experiments, it appears that the goal-recall component was not explicitly incorporated.
These points limit my ability to fully support acceptance of the paper.

**Limitations:**

Yes.

**Quality:**

2

**Strengths And Weaknesses:**

**Strengths**

- The work unifies the MEC–HPC circuit within a single-layer RNN, avoiding hand-crafted projection matrices between modules.
- The main claims are substantiated by both simulations and theoretical derivations.
- Ablation and comparison studies are provided, such as evaluations of path integration using GCs alone versus the full model, and recall performance with GC, SMC, and GC+SMC populations.

**Weaknesses**

- The novelty of the contribution is not clearly articulated. Many of the core results—such as grid cell path integration, emergence of place cells, and reconstruction from GC activity—have been previously explored in the literature.
- In Section 3, while the model demonstrates goal retrieval via sensory cues, this property does not appear to be integrated into the planning process. Instead, the planning submodule receives a separate, direct goal input, rather than leveraging retrieval from GCs.
- The training procedures for different model properties (path integration, memory retrieval, planning) seem to be conducted separately, rather than via a unified training protocol. Details regarding training procedures for each task are sometimes unclear.
- Minor issues:
  + Figures lack clarity; for example, the markers in Figure 2c are difficult to distinguish.
  + Not all notations are clearly defined.

---

> ### Author Rebuttal · Authors · 2025-07-31
>
> We thank the reviewer b4bC for their insightful comments which we address below.
>
> > **The novelty of the contribution**
>
> The main contribution of this paper is to answer the questions: "Why do animals maintain both place cells and grid cells?" and "What can animals do with these two representations to support internally motivated navigation tasks?" The answers to these questions form the core contributions of this work:
>
> - First, we propose that place cells serve to auto-associate the spatially modulated cells (SMCs) and grid cells (GCs), such that the two types of cells are tightly coupled within an environment and dynamically rewired across environments. Here, we do not intend to answer why place cells emerge, but rather to refine the functional role of place cells proposed in previous theories.
>
> - While this is an extension of previous place cell emergence theories, it makes predictions that cannot be captured by prior models, e.g., auto-associating SMCs and GCs allows retrieval of GC patterns at goal locations.
>
> - We subsequently show that using grid cells for planning is more efficient than direct planning with place cells or SMCs. Then, we explain why direct planning with place cells and SMCs, while feasible, may not be necessary. This auto-association also allows reconstruction of sensory experience from GC patterns, and we would like to highlight that this is not a previously known prediction and is a core contribution we aim to emphasize in this paper.
>
> - Finally, we extend the previous theoretical framework of grid-cell-based navigation (Bush et al., Using Grid Cells for Navigation, Neuron, 2015). Our extended framework provides a feasible implementation for both brain and ANNs without requiring an exponential number of neurons to encode pairwise displacements.
>
> > **Experiments on practical and real datasets**:
> We would also like to highlight our additional contribution that, as suggested by reviewer b2n6, we conducted experiments on realistic datasets using the Habitat Synthetic Scenes Dataset (HSSD) in Habitat-Sim, a visually realistic simulation environment.
>
> In our model, SMCs represent sensory observations. While the main text uses simulated sensory signals, here we replace them with features derived from visual observations from the Habitat Synthetic Scenes Dataset (HSSD): the agent captures images during exploration, which are passed through a vision encoder to generate SMC signals. We follow the same training procedure for HCMEC and planner models as in Section 4.4 and find that the model successfully reconstructs visual features from this realistic dataset.
>
> Importantly, our vision encoder produces features decodable back into images, allowing visualization of scenes along the planned path. All intermediate SMC states generated during planning can be decoded into images that closely match the expected scenes at their respective planned locations. We provide key implementation details below:
>
> - We discretize the environment into an $X_w \times X_h$ spatial grid and capture $W \times H$ allocentric panorama images at each grid point.
>
> - We fine-tuned a generalized visual encoder (a modified Masked AutoEncoder (MAE; He et al., 2021)) on images from this specific simulation gym. This MAE learns to encode images into compact feature representations, which can later be decoded back to the original images. Compactly, it can be expressed as $D(E(I)) \approx I$, where $I$ is the image, $E: \mathbb{R}^{W \times H} \to \mathbb{R}^{512}$ is the encoder, and $D: \mathbb{R}^{512} \to \mathbb{R}^{W \times H}$ is the decoder.
>
> - At each location, the image is encoded into a 512-dim vector used as the SMC response, enabling training of HCMEC and the planner as in the main text.
>
> - During planning, the HCMEC updates the SMC region to intermediate states ${z_0, \dots, z_T}$, where $z_i \in \mathbb{R}^{512}$. Decoding them with $D$ yields images ${I_0, \dots, I_T}$ that closely resemble the expected views at the corresponding planned locations.
>
> > **Integrating goal retrieval into the path-planning process**
>
> In the main text, we separated retrieval from planning to maintain a clearer separation of concerns across sections. Following the reviewer’s suggestion, we conducted an additional experiment that integrates the recall process directly into planning. We will include this additional result in the supplemental materials.
>
> > **Detailed training procedures**
>
> Supplemental Materials contain a detailed description of the simulation and training procedure. This is referred to on Line 127, 176, and 183. We will make this more prominent. We will also open-source our training code and model weights to ensure reproducibility.
>
> > **Figure clarity and defining all notations**
>
> Thanks. We will revise Fig 2. We would be grateful if you can point us to notations that were not defined, and fix the issue.
>
> > **On partial supervision of grid cells**
>
> Our core contributions and claims do not depend on co-emergence of grid cells, or having idealized grid cell responses.
>
> First, we are showing that if place cells (PCs) auto-associate grid cells and spatially modulated cells (SMCs), the agent can use sensory cues to recall GC patterns. Both recall and reconstruction of sensory experiences during planning work, as long as GC and SMC patterns are spatially stable.
>
> We will modify Section 4 to include a short mathematical argument that shows that planning with grid cells does not require idealized GC representations. It will consist of the following claims. Within a single grid-scale, the displacement vector can be correctly decoded even if the grid field undergoes shearing and compression (in 1D these two are directly related to grid scale, while the 2D case can be decomposed into 1D cases on non-colinear axes). Larger displacements across multiple grid-scales in sheared and compressed grid-fields can be pieced together by the decoding process. Decoding is also robust with sub-optimal grid-scales. The equation on Line 276 can be written in a more general form $$ \hat{d} = \frac{1}{2\pi m} \cdot \left( \sum_{i=1}^{k} \ell_{i} \cdot Z_i + \sum_{j=k+1}^{m} \ell_{j} \cdot \Delta \phi_i \right) $$Under optimal scaling, $\ell_i = \ell_{0} s^i$ but so long as there are sufficiently many scales, the predicted displacement is guaranteed to move the animal closer to the target.
>
> Unlike place cells **the grid cell patterns may not be learned but are believed to be dynamically self-organized via a network that was pre-wired by evolution**. For instance, genetically deleting the gene encoding a critical NMDA receptor subunit specifically disrupts grid patterns without affecting other spatial cell types (Gil et al., _Nat Neurosci._, 2018), which indicates that the underlying molecular components and the capacity to form grid-like patterns are at least partly encoded in the genome. Existing theoretical literature also provides evidence suggesting the grid cells may not be learned but are made by a pre-wired dynamical attractor network (Fuhs & Touretzky, J. Neurosci. 2006; Burak & Fiete, PLoS Comp. Bio., 2009; Gardner et al., Nature. 2022). The supervised grid cell region can be directly replaced with one of the existing computational and theoretical models (Cueva & Wei, ICLR, 2018; Kang & Balasubramian, eLife, 2019; Sorscher et al. NeurIPS, 2019) of grid cells without affecting our claims of recall and reconstruction during planning. Here we only use the grid cells as pattern generators that generate grid-like patterns (idealized or not), and ask, “if we have these patterns, could the animal use them for recall and planning”.
>
> > **Path integration mechanism and projection from speed and direction cells ...**
>
> We describe the path integration mechanism in Section 2.1.1. Specifically, speed and direction signals are directly input into designated speed and direction cells, which in turn project to grid cells through learned recurrent connections embedded within the single recurrent weight matrix (Eq. 2; Figure 1b, d). Grid cells are trained to infer the ground-truth GC responses from their initial state and the sequence of speed and direction signals relayed via these input cell types. While we did not explicitly analyze the emergence of conjunctive cells in our model, we observed band-like tuning patterns in the direction cells. Due to NeurIPS rebuttal constraints, we are unable to include these visualizations here but will provide them in the supplementary materials.
>
> > **Why is the SMC trained separately on masked sensory inputs?**
>
> The SMC is not trained separately on masked sensory inputs. We extend classic auto-association theories of place cells by proposing that they auto-associate not only SMCs but also grid cell patterns. This leads to two key predictions: sensory cues can recall GC patterns, and intermediate GC states during planning can be used to reconstruct sensory experiences.
>
> To trigger auto-association, the original signals must be distorted. Prior work (Wang et al., Time Makes Space, NeurIPS 2024) used masking for this purpose. In our model, SMCs simulate sensory input and GCs simulate path integration. Since these signals are not causally linked, we apply independently sampled masking ratios. The training process remains fully integrated, not separated into stages.
>
> > **Has the planner RNN been analyzed or dissected to confirm that it implements the proposed planning mechanism described in Sections 4.1–4.3?**
>
> The discussion in Section 4.1 and 4.3 is validated by the fact that we can decode back the location and trajectory of the animal correctly using SMCs and GCs. Section 4.2 provides a conceptual framework to think of how the RNN could generalize to planning for long trajectories after learning to plan for short trajectories. It is difficult to check the mechanism of the RNN, e.g., whether or not it instantiates a Markov chain that we have alluded to. We will preface the paragraph on Line 238 to emphasize this point.

---

> > ### Comment · Area_Chair_gtyS · 2025-08-05
> >
> > Dear Reviewer,
> >
> > The authors have already responded to your initial questions. As the deadline for the reviewer-author interaction session is approaching on August 6th, please begin addressing any further concerns or questions you may have. If you have no additional queries, kindly update your rating and submit your final decision.
> >
> > Thank you for your valuable contributions to NeurIPS.
> >
> > Best, AC

---

> > ### Comment · Reviewer_b4bC · 2025-08-05
> >
> > I thank the authors for their detailed response. I still have some concerns regarding the following points:
> >
> > > We propose that place cells serve to auto-associate the spatially modulated cells (SMCs) and grid cells (GCs)
> >
> > How does this point depart from the heteroassociative memory idea proposed in Chandra, S. et al., Nature, 2025?
> >
> > > auto-associating SMCs and GCs allows retrieval of GC patterns at goal locations
> >
> > If the goal-related GC pattern is only activated at goal location, how does it benefit planning at locations away from the goal?
> >
> > >On partial supervision of grid cells
> >
> > I want to clarify that I was not questioning either the emergence or capability of GCs. I was asking that why did you call it 'partial' when you have a full observation and groud truth label of GCs at every time steps.

---

> > > ### Author Response · Authors · 2025-08-05
> > >
> > > We thank the reviewer for their response and the chance to clarify our intent.
> > >
> > > > How does this point depart from the heteroassociative memory idea proposed in Chandra, S. et al. Nature 2025.
> > >
> > > Chandra et al., 2025. proposed a novel framework that can be succinctly summarized as: _if place cells, sensory input, and grid cells are wired together, the periodic structure of grid cell activity can increase memory capacity in the hippocampus by imposing a scaffold-like structure on the memory landscape_. Their work focuses on how this wiring can enhance memory capacity and support sequential memory.
> > >
> > > In contrast, our work addresses a different question: _how might a complete planning process be achieved using grid and place cell representations?_ We formulate the planning task as follows: the animal first recalls a location via sensory cues associated with the goal, then constructs a sequence of actions (i.e., displacement vectors) tracing a path toward that goal.
> > >
> > > While both works explore potential wiring regimes between grid cells, place cells, and sensory signals, we focus on how planning is possible through this wiring. Chandra et al. emphasize enhanced memory capacity and sequential memory; in that sense, our work can be seen as both an extension of and a parallel perspective to theirs. We highlight three key distinctions:
> > >
> > > 1. **Goal recall from sensory cues**:
> > >     We explain how associating grid cell activity with sensory patterns allows the animal to recall the grid representation of a goal location directly from sensory input. Chandra et al. primarily discuss how such associations increase the number of sensory patterns that can be stored, but do not explore whether grid representations can be recalled this way. Moreover, their framework assumes plasticity only between sensory and place cells, and therefore reconstruction of grid patterns from sensory cue might be challenging.
> > > 2. **Planning in grid cell space**:
> > >     We show why planning should be performed using grid cell patterns rather than place cell patterns: grid cells allow generalization across environments and are less context-dependent. Planning is not a focus of Chandra et al., and is not explored in their framework.
> > > 3. **Reconstructing sensory experience during planning**:
> > >     Even when planning occurs in grid space, we demonstrate how place cells enable reconstruction of intermediate sensory experiences along the planned path. Since Chandra et al. do not focus on planning, this forms our third primary contribution.
> > >
> > > Altogether, while our framework is distinct and novel in its focus and formulation, it can be regarded as a complementary extension of Chandra et al., emphasizing how the same underlying wiring logic might also support complex goal-directed planning. We will edit the text to clarify and expand these points concerning the relationship, and novelty relative to, Chandra et al.
> > >
> > > > If the goal-related GC pattern is only activated at the goal location, how does it benefit planning at locations away from the goal?
> > >
> > > The goal-related grid cell (GC) pattern is not activated only at the goal location. This is precisely what makes our proposed recall framework of interest. In classic path-integration theories, GC activity corresponds to the animal’s physical location. However, in our framework, once place cells have auto-associated sensory cues with GC patterns, any sensory cue linked to the goal can trigger retrieval of the goal GC pattern.
> > >
> > > For example, during exploration, place cells may form associations between the GC pattern active at a location and the sensory signals observed at that same place. Later, this association allows the animal to pattern-complete and retrieve the corresponding GC pattern upon encountering or even internally recalling those sensory cues, regardless of its physical position.
> > >
> > > Once the goal GC pattern is recalled, as we have shown, the animal can then plan a path to the goal and reconstruct the expected sensory experience along that planned route.
> > >
> > > > On partial supervision of grid cells
> > >
> > > We apologize for the confusion. We will revise the phrase _"partially supervised grid cells"_ to _"supervised grid cells that learned path integration"_ for clarity.
> > >
> > > To clarify our original intent: the speed, SMC, and direction cells in our network receive masked versions of their respective signals and are trained to match the unmasked versions, which is more akin to pattern completion. In contrast, the grid cells do not receive ground truth signals at any time step, masked or unmasked, except at initialization when $t = 0$. We originally used the term _"partial"_ to emphasize that grid cells are not simply performing pattern completion from ground truth but instead must learn to perform path integration. However, we agree that in standard machine learning terminology, this setup may be better described as supervision. We thank the reviewer for this suggestion and will revise the term accordingly.

---

### Official Review · Reviewer_b2n6 · 2025-07-03

**Clarity:** 3
**Significance:** 3
**Originality:** 2
**Rating:** 4
**Confidence:** 2

**Summary:**

This paper proposes a computational framework, REMI, to explain how animals might integrate hippocampal place cells (HPCs) and medial entorhinal grid cells (GCs) to support spatial memory and path planning. The authors hypothesize that GCs and PCs work together to enable animals to recall goal locations from sensory cues and plan paths, even through unvisited areas.

To test this, the authors develop a single-layer recurrent neural network (RNN) model that simulates the HC-MEC loop, incorporating sensory, speed, and direction inputs. GCs are partially supervised to emulate path integration, while HPCs emerge through recurrent dynamics. The network demonstrates that sensory cues at goal locations can trigger recall of corresponding GC states via HPCs. These GC representations can then be used for planning, with grid-space trajectories decoded into speed and direction commands. Furthermore, the model shows that intermediate GC states can drive the reconstruction of sensory experiences via HPCs during planning.

The paper also formalizes grid-space planning mechanisms, including decoding displacements using grid phase differences, learning local transition rules for sequential planning, and combining multiple spatial scales for more accurate long-range path planning.

**Questions:**

1. How does the proposed planning mechanism compare to other RL-based navigation models in terms of performance and efficiency?

2. The paper claims HPCs reconstruct sensory experiences during planning. Is there empirical evidence (e.g., firing patterns) to support this claim in the model?

3. How does performance (e.g., recall accuracy, planning success) vary with different levels of partial supervision or noise in the GC signals?

**Ethical Concerns:**

["NO or VERY MINOR ethics concerns only"]

**Final Justification:**

Thank the authors for the response, which addresses my concerns. I'll keep my positive rating unchanged.

**Limitations:**

yes

**Quality:**

3

**Strengths And Weaknesses:**

** Strengths

1. The use of a unified RNN architecture that simulates HC-MEC dynamics and planning within a biologically plausible framework is well-conceived and systematically developed.

2. The exposition is generally clear and logically structured, progressing from biological background and motivation to model design, experiments, and implications.

3. The paper addresses a central question in spatial cognition—why animals maintain dual spatial representations—and provides a mechanistic hypothesis with experimentally testable predictions. The framework may guide future research on memory-guided navigation and its neural implementation.

4. The integration of auto-associative memory recall with grid-cell-based planning is a novel and biologically plausible hypothesis. It extends prior work by showing how GC-based planning can generalize to unvisited locations and reconstruct sensory experiences.

** Weakness

1. The term "auto-association" is central but not rigorously defined in the context of the HC-MEC loop.

2. The proposed model lacks experiments on practical and real datasets.

---

> ### Author Rebuttal · Authors · 2025-07-31
>
> We thank reviewer b2n6 for their thoughtful review and respond below to the comments
>
> > **Definition of auto-association**
>
> We adopt the term from prior literature on the hippocampus's role in memory (Treves & Rolls, Hippocampus, 1994; ET. Rolls. Front. Syst. Neurosci. 2013.). Formally, we define an auto-association system as a function $f: \mathbb{R}^n \to \mathbb{R}^n$ such that $f(\mathbf{x}') = \mathbf{x}$ if $\mathbf{x}’ \approx \mathbf{x}$. In the scope of this paper, consider two vectors representing the Spatially Modulated Cells (SMCs) and Grid Cells (GCs), denoted as $\mathbf{x_{\text{smc}}}$ and $\mathbf{x_{\text{gc}}}$, respectively. These vectors, at any point during navigation, may be masked into $\mathbf{x_{\text{smc}}}'$ and $\mathbf{x_{\text{gc}}}'$ due to imperfect observation or path integration. We suggest that place cells serve as the function $f(\cdot)$ such that $f([\mathbf{x_{\text{smc}}}' ; \mathbf{x_{\text{gc}}}' ]) = [\mathbf{x_{\text{smc}}} ; \mathbf{x_{\text{gc}}} ]$.
>
> > **Experiments on practical and real datasets**
>
> We conducted additional experiments using the Habitat Synthetic Scenes Dataset (HSSD) in Habitat-Sim, a visually realistic simulation environment. In our model, SMCs represent sensory observations. While the main text uses simulated sensory signals, here we replace them with features derived from visual observations from the Habitat Synthetic Scenes Dataset (HSSD): the agent captures images during exploration, which are passed through a vision encoder to generate SMC signals. We follow the same training procedure for HCMEC and planner models as in Section 4.4 and find that the model successfully reconstructs visual features from this realistic dataset.
>
> Importantly, our vision encoder produces features decodable back into images, allowing visualization of scenes along the planned path. All intermediate SMC states generated during planning can be decoded into images that closely match the expected scenes at their respective planned locations. We provide key implementation details below:
>
> - We discretize the environment into an $X_w \times X_h$ spatial grid and capture $W \times H$ allocentric panorama images at each grid point.
>
> - We fine-tuned a generalized visual encoder (a modified Masked AutoEncoder (MAE; He et al., 2021)) on images from this specific simulation gym. This MAE learns to encode images into compact feature representations, which can later be decoded back to the original images. Compactly, it can be expressed as $D(E(I)) \approx I$, where $I$ is the image, $E: \mathbb{R}^{W \times H} \to \mathbb{R}\_+^{512}$ is the encoder, and $D: \mathbb{R}\_+^{512} \to \mathbb{R}^{W \times H}$ is the decoder.
>
> - At each location, the image is encoded into a 512-dim vector used as the SMC response, enabling training of HCMEC and the planner as in the main text.
>
> - During planning, the HCMEC updates the SMC region to intermediate states ${z_0, \dots, z_T}$, where $z_i \in \mathbb{R}^{512}$. Decoding them with $D$ yields images ${I_0, \dots, I_T}$ that closely resemble the expected views at the corresponding planned locations.
>
> >**Comparison to existing RL-based navigation methods**
>
> We intend this work primarily to suggest answers to the question of why animals maintain two systems to encode space. We propose that grid cell representations serve as efficient, generalizable encodings for planning across diverse environments, but they lack sensory detail and therefore require hippocampal place cells (HPCs) to pattern-complete the corresponding sensory information along the planned path. The most relevant metric for this study is the successful path-planning ratio; we presented an extensive test of the success planning ratio of the HCMEC model in the last section of this rebuttal.
>
> In the paper, to emphasize biological realism, we adopt a simplistic continuous-time RNN for modeling. However, the proposed framework is directly applicable to more advanced architectures such as GRUs or LSTMs, which may yield improved performance. We now additionally trained a variant with a simple layer normalization step before the hidden state update, which significantly increases the success rate to 0.99, making the approach potentially relevant for robotics applications.
>
> > **Empirical evidence supporting HPC reconstructs sensory experiences during planning**
>
> To address this, we first test whether hippocampal place cells (HPCs) specifically contribute to trajectory reconstruction. This can be evaluated by checking whether intermediate HPC states during planning decode into a valid trajectory. For each planning trial, we record all HPC states and decode each time step into a location using nearest neighbor search over ratemaps aggregated during testing.
>
> While the main text focuses on decoding using spatially modulated cells (SMCs) and grid cells, here we compare planning performance across 4096 trials using HPC-decoded versus SMC-decoded trajectories. We find that HPC-decoded trajectories also reliably reach the goal:
>
> | Cell Type | Distance to Goal (cm) | Success Rate |
> | ------------- | ---------------- | ------------ |
> | SMCs | 4.27 ± 5.22 | 0.96 |
> | HPCs | 4.97 ± 7.80 | 0.94 |
>
> A complementary perspective is to test whether the reconstructed experience can be decoded beyond nearest neighbor search. The experiment in Experiments on practical and real datasets provides strong support: we train a compact image encoder-decoder, replace SMC signals with encoded image features, and find that the reconstructed sensory signals along the planned path can be decoded back into meaningful images.
>
> > **Performance varies with different levels of partial supervision or noise**
>
> We conducted additional experiments to systematically distort the simulated ground truth grid cells. We trained 18 HCMEC and corresponding planner models (10 to evaluate the impact of noise and 8 to assess geometric distortions). Each model was evaluated on 4,096 planning trials. A trial is considered successful if the agent's final position is within 10 cm of the goal. Success rate is computed over all trials. On average, the goal is 57.54 ± 25.53 cm away from the start.
>
> For the noise experiments, we added post-activation Gaussian noise (standard deviation 0.1 to 1.0) to all hidden units of the HCMEC at each time step during training. No noise was added at test time (planning phase).
>
> Due to space constraints, we report results based on decoding from SMCs, though we verified that results are consistent when decoding from GCs. In all cases, the mean distance to the goal remains below 10 cm (in a 100 cm × 100 cm environment), and success rates remain high across noise levels. This suggests that the planner can effectively operate even with noisy GC representations.
>
> | Noise (std) | Distance to Goal (cm) | Success Rate | Detour Ratio |
> | ------------- | ---------------- | ------------ | ------------ |
> | 0.1           | 2.20 ± 5.89      | 0.99         | 1.66         |
> | 0.2           | 2.35 ± 5.68      | 0.99         | 2.00         |
> | 0.3           | 1.82 ± 5.09      | 0.99         | 1.88         |
> | 0.4           | 3.75 ± 8.84      | 0.98         | 2.10         |
> | 0.5           | 2.62 ± 3.43      | 0.98         | 2.36         |
> | 0.6           | 3.12 ± 3.72      | 0.97         | 2.33         |
> | 0.7           | 3.31 ± 5.66      | 0.99         | 2.55         |
> | 0.8           | 2.98 ± 2.78      | 0.99         | 2.60         |
> | 0.9           | 3.00 ± 3.99      | 0.98         | 2.31         |
> | 1.0           | 5.30 ± 6.98      | 0.91         | 2.70         |
>
> Next, we added noise to the ground-truth GC signal that is used at the first time step during planning.
>
> | Noise (std) | Distance to Goal (cm) | Success Rate | Detour Ratio |
> | ------------- | ---------------- | ------------ | ------------ |
> | 0.1           | 2.45 ± 2.49      | 0.99         | 2.16         |
> | 0.2           | 2.40 ± 4.37      | 0.99         | 1.91         |
> | 0.3           | 1.89 ± 5.41      | 0.99         | 1.98         |
> | 0.4           | 4.10 ± 7.66      | 0.96         | 2.54         |
> | 0.5           | 4.36 ± 5.01      | 0.95         | 3.28         |
> | 0.6           | 4.05 ± 5.48      | 0.96         | 2.44         |
> | 0.7           | 4.01 ± 6.70      | 0.97         | 2.53         |
> | 0.8           | 4.79 ± 6.00      | 0.91         | 2.63         |
> | 0.9           | 5.54 ± 8.43      | 0.91         | 2.67         |
> | 1.0           | 7.72 ± 13.24    | 0.85         | 2.49         |
>
> Additionally, we trained another 8 models without noise but with sheared grid fields.
>
> | Shearing Angle | Residual to Goal (cm) | Success Rate | Detour Ratio |
> | -------------- | --------------------- | ------------ | ------------ |
> | 40             | 9.53 ± 16.39          | 0.86         | 2.45         |
> | 45             | 7.88 ± 17.66          | 0.91         | 2.10         |
> | 50             | 7.40 ± 18.18          | 0.91         | 2.12         |
> | 55             | 5.48 ± 11.38          | 0.92         | 2.38         |
> | 60             | 7.66 ± 16.29          | 0.87         | 2.26         |
> | 65             | 2.47 ± 7.13           | 0.98         | 2.04         |
> | 70             | 5.13 ± 10.22          | 0.91         | 2.44         |
> | 75             | 6.38 ± 14.26          | 0.91         | 2.12         |
> | 80             | 4.46 ± 11.31          | 0.94         | 1.91         |

---

> > ### Comment · Reviewer_b2n6 · 2025-08-04
> >
> > Thank the authors for the response. I'll keep my positive rating unchanged.

---

### Official Review · Reviewer_u9fY · 2025-07-04

**Clarity:** 4
**Significance:** 3
**Originality:** 3
**Rating:** 5
**Confidence:** 4

**Summary:**

This paper proposes REMI, a framework explaining how animals use both grid cells and place cells for spatial navigation and path planning. The authors suggest that place cells auto-associate sensory information with grid cell patterns, enabling animals to recall goal locations from sensory cues and then plan paths using stable grid cell representations. They demonstrate this with a model that shows how grid cells can support sequential planning through local transition rules that generalize to long-distance navigation, while place cells can reconstruct sensory experiences along planned paths.

**Questions:**

Any arbitrarily complicated recurrent structure can be written as a single layer RNN. Since you train the HC-MEC and planning networks in very different ways and at different times, why does it make sense to think of them as the 'same' RNN?
The paper suggests a connection to successor features, which at a high-level seems plausible. Is there a rigorous connection that can be specified demonstrating that the same future state transition probabilities are encoded in the trained model?
How accurate is your actual path integration? Do you have quantitative metrics showing this for different time scales? How does it compare to other neural path integrators in terms of accuracy?

**Ethical Concerns:**

["NO or VERY MINOR ethics concerns only"]

**Final Justification:**

Running the model on the new dataset is a nice addition. I don't have any other questions and appreciate the responses to my original queries.

**Limitations:**

yes

**Quality:**

3

**Strengths And Weaknesses:**

The work provides a compelling theoretical framework that unifies the functional roles of grid cells and place cells, addressing the important question of why animals maintain two distinct spatial representation systems. The authors successfully model the HC-MEC loop in a single RNN that simultaneously exhibits grid cell path integration, place cell emergence, and planning capabilities. It was unclear if the planning network is properly thought of as part of the same single layer RNN, as it was added on afterwards, but regardless it is an interesting model. The work makes concrete, testable experimental predictions about how sensory cues should reactivate grid cell patterns and how selective brain region disruptions should affect navigation, which provide important links to empirical work.

A significant limitation is the reliance on partially supervised grid cells rather than demonstrating their natural co-emergence with place cells, which somewhat undermines the biological plausibility claims and leaves questions about how this system would actually develop in real neural circuits. However, this limitation is acknowledge by the authors and leaving it to future work seems appropriate given the strong contributions of the remainder.

---

> ### Author Rebuttal · Authors · 2025-07-31
>
> We thank reviewer u9fY for their thoughtful review and the endorsement of our work. We are excited to see that you think this is a compelling theoretical framework that unifies the functional roles of grid cells and place cells. As you have mentioned, we indeed attempt to propose this framework to construct a system-level theory to explain the empirical works on each neural sub-region, as collectively these patterns must be preserved by evolution due to their functional importance.
>
> > **Experiments on practical and real datasets**: We would first like to highlight an additional contribution: as suggested by reviewer b2n6, we conducted experiments in a visually realistic simulation environment (Habitat-Sim) using realistic datasets.
>
> In our model, SMCs represent sensory observations. While the main text uses simulated sensory signals, here we replace them with features derived from visual observations from the Habitat Synthetic Scenes Dataset (HSSD): the agent captures images during exploration, which are passed through a vision encoder to generate SMC signals. We follow the same training procedure for HCMEC and planner models as in Section 4.4 and find that the model successfully reconstructs visual features from this realistic dataset.
>
> Importantly, our vision encoder produces features decodable back into images, allowing visualization of scenes along the planned path. All intermediate SMC states generated during planning can be decoded into images that closely match the expected scenes at their respective planned locations. We provide key implementation details below:
>
> - We discretize the environment into an $X_w \times X_h$ spatial grid and capture $W \times H$ allocentric panorama images at each grid point.
>
> - We fine-tuned a generalized visual encoder (a modified Masked AutoEncoder (MAE; He et al., 2021)) on images from this specific simulation gym. This MAE learns to encode images into compact feature representations, which can later be decoded back to the original images. Compactly, it can be expressed as $D(E(I)) \approx I$, where $I$ is the image, $E: \mathbb{R}^{W \times H} \to \mathbb{R}^{512}$ is the encoder, and $D: \mathbb{R}^{512} \to \mathbb{R}^{W \times H}$ is the decoder.
>
> - At each location, the image is encoded into a 512-dim vector used as the SMC response, enabling training of HCMEC and the planner as in the main text.
>
> - During planning, the HCMEC updates the SMC region to intermediate states ${z_0, \dots, z_T}$, where $z_i \in \mathbb{R}^{512}$. Decoding them with $D$ yields images ${I_0, \dots, I_T}$ closely resemble the expected views at the corresponding planned locations.
>
> **Here we address the questions in the review:**
>
> > **It was unclear if the planning network is thought of as part of the same single-layer RNN:**
>
> We train the planning and HCMEC networks in a two-stage setting primarily because they may correspond to two separate behavioral processes. HCMEC training models the phase when the animal explores the environment and builds representations of it, whereas the planner models how, once a spatial representation is formed, the animal might virtually navigate on it to plan paths. At the implementation level, however, this is equivalent to initializing a single network with both HCMEC and planner units, then alternating between training to encode the space and to plan for navigation.
>
> Additionally, we used the term "single layer" to emphasize that in the HCMEC model, both HC and MEC can project to each other, unlike prior models that typically assume unidirectional information flow. As the reviewer noted, we are leveraging the fact that any arbitrarily complex RNN can be represented as a single-layer RNN. This allows us to model bidirectional interactions between multiple neural subpopulations using a single recurrent weight matrix. These interactions are entirely determined by the optimization objective, without the need to define or handcraft structures.
>
> > **On partial supervision of grid cells**
>
> Our core contributions and claims do not depend on co-emergence of grid cells, or having idealized grid cell responses.
>
> First, we are showing that if place cells (PCs) auto-associate grid cells and spatially modulated cells (SMCs), the agent can use sensory cues to recall GC patterns. Both recall and reconstruction of sensory experiences during planning work, as long as GC and SMC patterns are spatially stable.
>
> We will modify Section 4 to include a short mathematical argument that shows that planning with grid cells does not require idealized GC representations. It will consist of the following claims. Within a single grid-scale, the displacement vector can be correctly decoded even if the grid field undergoes shearing and compression (in 1D these two are directly related to grid scale, while the 2D case can be decomposed into 1D cases on non-colinear axes). Larger displacements across multiple grid-scales in sheared and compressed grid-fields can be pieced together by the decoding process. Decoding is also robust with sub-optimal grid-scales. The equation on Line 276 can be written in a more general form $$ \hat{d} = \frac{1}{2\pi m} \cdot \left( \sum_{i=1}^{k} \ell_{i} \cdot Z_i + \sum_{j=k+1}^{m} \ell_{j} \cdot \Delta \phi_i \right) $$Under optimal scaling, $\ell_i = \ell_{0} s^i$ but so long as there are sufficiently many scales, the predicted displacement is guaranteed to move the animal closer to the target.
>
> Unlike place cells **the grid cell patterns may not be learned but are believed to be dynamically self-organized via a network that was pre-wired by evolution**. For instance, genetically deleting the gene encoding a critical NMDA receptor subunit specifically disrupts grid patterns without affecting other spatial cell types (Gil et al., _Nat Neurosci._, 2018), which indicates that the underlying molecular components and the capacity to form grid-like patterns are at least partly encoded in the genome. Existing theoretical literature also provides evidence suggesting the grid cells may not be learned but are made by a pre-wired dynamical attractor network (Fuhs & Touretzky, J. Neurosci. 2006; Burak & Fiete, PLoS Comp. Bio., 2009; Gardner et al., Nature. 2022). The supervised grid cell region can be directly replaced with one of the existing computational and theoretical models (Cueva & Wei, ICLR, 2018; Kang & Balasubramian, eLife, 2019; Sorscher et al. NeurIPS, 2019) of grid cells without affecting our claims of recall and reconstruction during planning. Here we only use the grid cells as pattern generators that generate grid-like patterns (idealized or not), and ask, “if we have these patterns, could the animal use them for recall and planning”.
>
> > **Connection to Successor Features**
>
> Here, we analyze why, at the implementation level, our network must encode future state. In REMI, HPCs must predict both GC and SMC responses at the next timestep in order to pattern-complete. Since HPCs are direction-agnostic, accurately reconstructing GC and SMC activity requires encoding possible state transitions. Because adjacent locations tend to have higher transition probability (Stachenfeld et al., 2017), our model implicitly reflects the successor representation framework of HPCs, where transition structure is embedded in the pattern-completion dynamics.
>
> Regarding our discussion of grid cell transition probabilities, we primarily use it as a conceptual framework to understand how the RNN might generalize from planning short trajectories to longer ones, and why such generalization is only feasible with the periodic grid structure. It is difficult to check the mechanism of the RNN, e.g., whether or not it instantiates a Markov chain that we have alluded to. We will preface the paragraph on Line 238 to emphasize this point.
>
> > **Path Integration Accuracy**
>
> We tested the path integration accuracy of models with (Figure 1d) and without (Figure 1b) Spatially Modulated Cells (SMCs). Both models contain the same number of Grid, Direction, and Speed Cells, with the full HCMEC model additionally including SMCs and Place Cells. The HCMEC model was trained with a masking ratio for each trial sampled uniformly from $\mathcal{U}[0, 1]$.
>
> To test path integration, the GC subregion was initialized with the ground truth grid state. During testing, the model received unmasked speed and direction inputs. In the full HCMEC model, the SMC subregion additionally received SMC signals. Grid cell subregions were expected to update their internal grid states accordingly.
>
> We evaluated each model over 100 seconds across 512 independent trials. GC responses were decoded into spatial coordinates using nearest neighbor search, and errors were computed as Euclidean distances (cm) from the ground truth at [1, 5, 10, 20, 50, 100] seconds in a 100 cm × 100 cm environment (1cm/spatial pixel):
>
> - Full HCMEC model: [1.96 ± 1.58, 2.63 ± 3.26, 3.14 ± 6.32, 3.76 ± 8.23, 7.67 ± 17.06, 11.43 ± 22.47] (in cm)
> - Grid cell only model: [1.89 ± 2.92, 2.76 ± 5.34, 3.37 ± 7.11, 4.86 ± 10.77, 5.17 ± 12.39, 5.28 ± 12.74] (in cm)
>
> Both models show high path-integration accuracy over time. The error may result from the discretization of space and time in our simulation, which introduces small inaccuracies that accumulate during path integration. In contrast, biological systems may use additional signals such as boundary cells for error correction (Eli et al. Dynamic self-organized error-correction of grid cells. 2018). Future work may incorporate such signals to improve accuracy. The full HCMEC model shows slightly lower path-integration accuracy than the GC-only model, likely due to training with masked velocity, speed, and direction inputs.
>
> All experiments were conducted with agents moving at speeds sampled from a right-skewed log-normal distribution with mean and standard deviation of 10 cm/s. Since speed affects the temporal resolution of input signals, our results are robust across different timescales.

---

> > ### Comment · Area_Chair_gtyS · 2025-08-05
> >
> > Dear Reviewer,
> >
> > The authors have already responded to your initial questions. As the deadline for the reviewer-author interaction session is approaching on August 6th, please begin addressing any further concerns or questions you may have. If you have no additional queries, kindly update your rating and submit your final decision.
> >
> > Thank you for your valuable contributions to NeurIPS.
> >
> > Best, AC

---

### Note · Authors · 2025-08-12

We present REMI, a unified, system-level theory explaining how the hippocampus (HC) and medial entorhinal cortex (MEC) may support cue-triggered goal retrieval, path planning, and reconstruction of sensory experience along a planned route. We propose that place cells (PCs) may auto-associate grid cell (GC) states and sensory patterns during navigation, thereby encoding space more robustly than either system alone. This association naturally enables reconstruction of GC states from sensory input and vice versa, providing a mechanistic basis for these planning behaviors.

Strengths noted by the reviewers:
- **Reviewers uf97, b2n6, and KTLJ identified our foremost strength as providing this system-level theory of HC and MEC for planning while demonstrating the model’s planning capabilities in a single RNN.**
- All reviewers commended our unified MEC–HC–planning circuit in a single-layer RNN, avoiding handcrafted network structures.
- Reviewers uf97, b2n6, and KTLJ also highlighted that the technical implementation of REMI is well conceived.

Concerns and suggestions addressed during rebuttal:

- Supervision of grid cells: Reviewers uf97 and KTLJ asked whether our model relies on supervised grid cells. We addressed this from two perspectives: (i) our claims do not depend on GC co-emergence or idealized responses. Following KTLJ’s suggestion, we tested 9 models with sheared GC fields and 20 with varying noise; path planning performance remained robust to shear and declined only gradually with noise. (ii) GC patterns may arise from evolutionarily prewired circuits rather than learning. We showed that the GC module can be replaced by a dynamical attractor model or a trained RNN without affecting our theory.
- Comparison with successor representation (SR): Reviewers uf97 and KTLJ requested additional comparison with SR. We show that in REMI, for PCs to reconstruct MEC activity they must encode possible state transitions (see our response to KTLJ), thereby implicitly implementing SR
- Real world datasets: In response to reviewer b2n6, we replaced simulated sensory signals with images from an agent in a visually realistic simulator. We encoded images with an autoencoder and used the features as the sensory signal. During planning, GC patterns activated the sensory subregion; decoding it produced meaningful images along the planned path.
- Additional experiments (robustness, path integration acc, etc) requested by reviewers will all be included in revised manuscript.

---

### Decision · Program_Chairs · 2025-09-17

**Decision:**

Accept (poster)

**Comment:**

The paper introduces REMI, a unified, system-level theory for cue-triggered goal retrieval, path planning, and reconstruction of sensory experience along a planned route. Before the rebuttal, reviewers raised concerns regarding the lack of real-world datasets, unclear definition of “partially supervised,” noise robustness, and missing baseline comparisons. After the rebuttal, three reviewers recommended acceptance, while one insisted on rejection, primarily due to the lack of comparison with Chandra et al., Nature 2025. Although the authors provided a strong rebuttal, the reviewer remained unconvinced.

After reviewing the exchange, the AC agrees with the authors that the focus and model designs of the two papers are different and complementary.

Given that the merits of the work outweigh its limitations, the AC recommends acceptance (poster). The authors are strongly encouraged to incorporate the reviewers’ comments into the final version.